# THE ROLE OF DIVERSITY IN IN-CONTEXT LEARNING FOR LARGE LANGUAGE MODELS

## ABSTRACT

In-context learning (ICL) is a crucial capability of current large language models (LLMs), where the selection of examples plays a key role in performance. While most existing approaches focus on selecting the most *similar* examples to the query, the impact of *diversity* in example selection remains underexplored. We systematically investigate the role of *diversity* in in-context example selection through experiments across a range of tasks, from sentiment classification to more challenging math and code problems. Experiments on Llama-3.1, Gemma-2, and Mistral-v0.3 families of models show that diversity-aware selection methods improve performance, particularly on complex tasks like math and code, and enhance robustness to out-of-distribution queries. To support these findings, we introduce a theoretical framework that explains the benefits of incorporating diversity in in-context example selection.

## 1 INTRODUCTION

In-context learning (ICL) (Brown et al., 2020) has emerged as one of the most significant and versatile capabilities of large language models (LLMs). This paradigm allows a model to adapt to a vast array of new tasks on the fly, simply by conditioning on a prompt containing a few input-output examples, all without requiring updates to its parameters. The power and resource efficiency of ICL have made it a cornerstone of LLM applications, ranging from simple text classification(Min et al., 2022) and commonsense reasoning (Srivastava et al., 2023) to complex, multi-step tasks like mathematical problem-solving(Wei et al., 2022) and code generation (Chen et al., 2021).

The effectiveness of ICL, however, is highly sensitive to the choice of in-context examples (Lu et al., 2021; Liu et al., 2021; Chang and Jia, 2023). This makes example selection a critical area of study. To address this, prior work has explored various selection strategies: choosing examples most *similar* to the query in embedding space (Liu et al., 2021; Yang et al., 2022; Wu et al., 2023; Qin et al., 2023), maximizing feature *coverage* (Levy et al., 2023; Ye et al., 2023; Gupta et al., 2023), selecting based on *difficulty* (Ma et al., 2025; Swayamdipta et al., 2020; Yuan et al., 2025; Cook et al., 2025), or choosing examples based on *sensitivity* (Chen et al., 2023). Other approaches train deep neural retrievers (Karpukhin et al., 2020; Rubin et al., 2022; Luo et al., 2023; Scarlatos and Lan, 2023) or leverage feedback from large language models to guide selection (Li and Qiu, 2023a; Chen et al., 2023; Wang et al., 2023). More discussion can be found in Appendix A. Among these, similarity-based methods remain the fundamental baseline due to their conceptual simplicity and consistent empirical success. However, relying solely on similarity can lead to redundancy among demonstrations and potentially omit important but less similar features (Levy et al., 2023; Gupta et al., 2023).

Within machine learning, *diversity* is also a fundamental principle for building robust and generalizable models, and its importance is widely recognized in related domains—such as fixed-prompt ICL with global demonstration sets (Li and Qiu, 2023b; Luo et al., 2024), active learning (Giouroukis et al., 2025; Shi and Shen, 2016), coreset construction (Wan et al., 2024; Zhan et al., 2025; Sener and Savarese, 2018), and instruction tuning (Wang et al., 2024). By exposing a model to a varied set of examples, we can prevent overfitting and encourage the learning of more abstract, transferable patterns. Given its foundational role, a deep understanding of diversity is crucial for unlocking the full potential of in-context learning.

Despite its importance, the explicit role of diversity in retrieval-based ICL remains underexplored. While some recent work has incorporated feature coverage as a proxy for diversity (Levy et al., 2023;

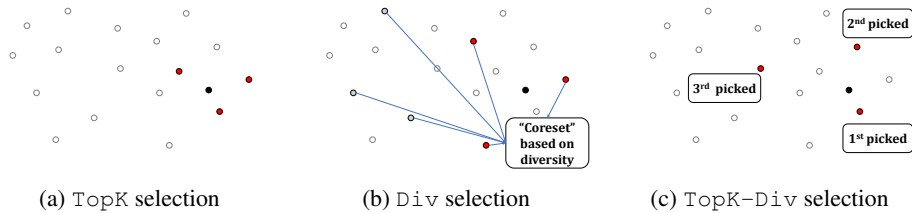

(a) `TopK` selection    (b) `Div` selection    (c) `TopK-Div` selection

Figure 1: An illustrative example for `TopK`, `Div`, and `TopK-Div` methods. Point filled in black denotes the query. **(a)** `TopK`: Select the most similar demonstrations (3 points filled in red) in the embedding space. **(b)** `Div`: First select a "coreset" based on some *diversity* metric, which is fixed for all queries (6 points filled in gray or red). Then select the most similar demonstrations (3 points filled in red) among this "coreset". **(c)** `TopK-Div`: Select the demonstrations sequentially based on the linear combination of similarity to the query and the diversity with the selected examples. The first example is the one most similar to the query. When picking the second, it is balanced between the similarity to the query and the diversity from the first example.

Ye et al., 2023; Gupta et al., 2023), this approach is limited in scope. Coverage primarily aims to span the concrete input features of a given query, which is a narrower goal than promoting the broader representational variety that is central to true diversity; see also examples in Appendix E that compare coverage with diversity. Other diversity-aware approaches have also been proposed, such as the S3 method from Kumari et al. (2024). However, that study was confined to simple classification and selection tasks. Consequently, the effectiveness of diversity in more complex, reasoning-based ICL applications remains an open question.

Furthermore, pursuing diversity without care can be counterproductive, as explicit diversity-aware selection risks retrieving examples that are too dissimilar from the query, potentially harming performance (An et al., 2023a). The field, therefore, lacks a systematic understanding of this trade-off. It remains unclear whether and when explicit diversity is beneficial—especially for tasks that lack clear local structure and demand more abstract reasoning. This gap in knowledge motivates the following fundamental questions:

> *Should we explicitly consider diversity when selecting in-context examples? If so, under what conditions does it outperform similarity-based methods? And fundamentally, why does diversity help?*

## 1.1 OUR CONTRIBUTIONS

We present, to the best of our knowledge, the first systematic investigation of the role of diversity in in-context learning. Our study spans a broad range of tasks—including sentiment classification, commonsense reasoning, math generation, reading comprehension, and SQL code generation—covering diverse types and varying levels of difficulty. We compare five demonstration selection methods: (1) random selection (`Rand`); (2) selecting the $K$ most similar examples to the query (`TopK`) (Liu et al., 2021); (3) Select the most representative examples from a similarity-reduced subset (`Div-S3`) (Kumari et al., 2024); (4) selecting similar examples from a diversity-reduced subset (`Div`) (Su et al., 2023), which relates to DPP-based diversity (Chen et al., 2018); and (5) a sequential method that balances similarity to the query and diversity among selected examples (`TopK-Div`). We are the first to systematically evaluate methods for (4) and (5) in the ICL setting. Their approaches are particularly compelling because they offer explicit control over the diversity level, enabling a tunable trade-off between selecting highly relevant examples and avoiding redundancy—a key factor in optimizing LLM performance within limited context windows. See Figure 1 for illustration and Section 2 for formal definitions.

Through experiments on frontier open-source models (Llama-3.1 (Dubey et al., 2024), Gemma-2 (Team et al., 2024), and Mistral-v0.3 (Jiang et al., 2023)), we reach the following findings.

**Finding 1**: *Diversity-aware demonstration selection methods achieve better performance on more "challenging" tasks like reading comprehension, math, and code.* As task difficulty increases, diversity-aware methods yield greater relative benefits, narrowing the gap with the `TopK` method or even surpassing it.

While changing the tasks and even the language model to use will change the ranking of the demonstration selection methods we test, in general we find that diversity-aware methods, namely `Div`, `TopK-Div` and `Div-S3`, perform better on more challenging tasks like reading comprehension, math, and simple code generation. On the other hand, on simple tasks like sentiment classification and multiple-choice, `TopK` performs the best (Table 1). TopK-Div achieves, on average, more than a 1% improvement over TopK on difficult tasks, whereas on simpler tasks TopK holds a marginal 0.14% average advantage over TopK-Div. Quantitative analysis of performance improvements under varying levels of added diversity for `TopK-Div` and `Div` demonstrates that more challenging tasks benefit more from increased diversity, further validating this finding (Figure 4, Table 7).

**Finding 2**: *Diversity-aware methods work better for out-of-distribution queries.* When the query and demonstrations come from different distributions, diversity-aware methods are more likely to perform well. For example, on sentiment classification, when both demonstration and query come from the SST-2 dataset, which consists of movie reviews, `TopK` performs the best, and there is a gap with all other methods (Table 2). The average performance gap between `TopK` and `TopK-Div` is 1.26% across the two models. However, when changing the demonstrations from SST-2 to IMDB (which also consists of movie reviews), `TopK-Div` outperforms `TopK` by 0.6% on average (Table 2). A similar observation holds for various splits of Geoquery dataset (Figure 2).

**Finding 3**: In the same task, diversity-aware methods likely perform better on "harder" examples, e.g. reading comprehension with longer context, or SQL code generation with more structures (Table 3). On the easier samples from GeoQuery and SQuAD, `TopK-Div` achieves an average accuracy improvement of 2.12% over `TopK`. On the more challenging samples, the average improvement of `TopK-Div` over `TopK` increases to 6.47%. Our analysis also reveals that diversity exhibits a "beyond-coverage" phenomenon, both at the task level and the example level (see discussion in Section 3.1 and Appendix E).

We discuss these findings in detail in Section 3. In addition, we conduct ablation studies across model scales (ranging from 1B to 70B parameters) and varying numbers of in-context demonstrations to examine the robustness of our conclusions. Beyond empirical trends, we extract practical insights from our study: our results offer actionable guidance for tuning the diversity level of demonstrations depending on the task type, such as reasoning, generation, or classification. Overall, our findings deepen the understanding of how diversity influences in-context learning, and inform principled strategies for demonstration selection in real-world applications.

## 2 BACKGROUND AND NOTATIONS

In this section, we introduce the in-context learning (ICL) paradigm, relevant demonstration selection methods, and associated notations.

**In-context learning (ICL).** A task $\mathcal{T} = (\mathcal{X}, \mathcal{Y}, P(y|x))$ defines a probabilistic mapping from an input $x \in \mathcal{X}$ to an output $y \in \mathcal{Y}$. For example, the task can be sentiment classification where the input space contains reviews of products and the output space contains the customer's corresponding sentiment (positive or negative). We are provided with a demonstration set $D = \{(x_i, y_i)\}_{i=1}^n$, where inputs $x_i$ are drawn from a demonstration distribution $\mathcal{D}_\mathcal{X}$ and $y_i \sim P_\mathcal{T}(y|x_i)$. Queries $x_q$ are drawn from a query distribution $\mathcal{Q}_\mathcal{X}$, which may differ from $\mathcal{D}_\mathcal{X}$ (representing shifts in domain or complexity). For math tasks, the demonstration set may contain many elementary-level problems, while the query may require solving more advanced, middle-school-level problems. Now given a query input $x_q \sim \mathcal{Q}_\mathcal{X}$, the in-context learning paradigm refers to the following capability of a large language model.

**Definition 2.1 (In-Context learning (ICL)).** Given an LLM, a prompting strategy `Prompt`, a demonstration set $D = \{(x_i, y_i)\}_{i=1}^n$, and a query $x_q$, ICL involves selecting a small subset $S = \{(x_{j_i}, y_{j_i})\}_{i=1}^K$ with shots $K$ from the demonstrations $D$. The LLM then predicts the output $y_q$ for $x_q$ as: $P_\mathcal{T}(y|x_q) \approx \text{LLM}(\text{Prompt}(S, x_q))$.

**Demonstration selection for ICL.** Choosing a small subset $S$ (see Theorem 2.1) is vital due to LLM context limits, efficiency needs, and the observation that excessive demonstrations can impair performance. Prior work has shown that ICL performance is highly sensitive to this selection (Liu et al., 2021), and thus sparks the study for *demonstration selection*. While numerous selection strategies are proposed , the most notable and effective methods are the ones that select the demonstrations most

similar to the query in the embedding space. Efforts are also made to retrieve the demonstrations using another model (can be another LLM), as well as considering diversity/coverage. However, there is no consensus on which method to use in a specific setting, and there is nearly no understanding of these methods (further discussed in Appendix A).

To analyze these methods and the role of diversity, we focus on five representative selection strategies:

**Method 1: `Rand`.** For a query $x_q$, this method uniformly and randomly selects $K$ demonstrations from the set $D$. Note that `Rand` can also be viewed as a method that is aware of diversity, but it has nothing to do with the coverage.

**Method 2: `TopK`.** This method selects $K$ demonstrations from $D$ that exhibit the highest cosine similarity to the query $x_q$ within an embedding space mapped by $E : \mathcal{X} \to \mathcal{E}$. It maximizes

$$\texttt{Similarity}(E(x_i), E(x_q)) := \frac{\langle E(x_i), E(x_q) \rangle}{\|E(x_i)\| \cdot \|E(x_q)\|}. \tag{1}$$

**Method 3: `Div-S3`.** This method, proposed by Kumari et al. (2024) for in-context demonstration selection, combines a similarity-based pruning step with a greedy submodular optimization to select examples that are both relevant and diverse. The approach aims to ensure representative coverage while maintaining closeness to the query. Although submodular diversity techniques have been well-studied in classical data selection (Lin and Bilmes, 2011; Prasad et al., 2014), their application in ICL has not been systematically explored.

**Method 4: `Div`.** This approach first constructs a diverse "coreset" $D_r \subset D$ of size $m$ (where $K \leq m \leq n$). Starting with one randomly chosen demonstration, $D_r$ is built greedily by adding $(x, y) \in D \setminus D_r$ that maximizes

$$\texttt{Diversity}(E(x), D_r) := 1 - \frac{1}{|D_r|} \sum_{(x_j, y_j) \in D_r} \texttt{Similarity}(E(x), E(x_j)), \tag{2}$$

we stop after $D_r$ contains $m$ examples. This is the procedure to get a diverse set of demonstrations for a task (Su et al., 2023). Subsequently, `TopK` selection is applied to $D_r$ to choose $K$ demonstrations for the query $x_q$. The coreset size $m$ controls the trade-off between diversity and similarity: setting $m = K$ emphasizes diversity by forcing selection from a small pool, while increasing $m$ shifts the method closer to `TopK` by enlarging the candidate set based on similarity.

**Method 5: `TopK-Div`.** This method serves as a combination of `TopK` and `Div`, which includes some awareness of the diversity through similarity-based selection. It is also a greedy-like procedure when selecting the demonstration set $S$. Suppose that $S$ does not reach size $K$, then we select the demonstration $(x, y) \in D \setminus K$ that maximize the following metric:

$$\alpha \cdot \texttt{Similarity}(E(x), e_q) + (1 - \alpha) \cdot \texttt{Diversity}(E(x), S), \tag{3}$$

The hyperparameter $\alpha$ governs the balance between diversity and similarity: setting $\alpha = 0$ emphasizes diversity among selected examples, while $\alpha = 1$ recovers the `TopK` method that prioritizes similarity to the query. We stop when $S$ has size $K$. For the first demonstration (when $S$ is empty), $\texttt{Diversity}(E(x), S)$ is defined as $0$, thus prioritizing similarity.

The use of `TopK-Div` and `Div` methods for demonstration selection is, to our knowledge, new in the ICL setting. Their flexibility in adjusting the diversity level offers practical value, as it enables task-specific tuning to improve performance; see Section 3.1 for details.

## 3 EXPERIMENTS AND FINDINGS

This section empirically tests whether diversity-aware retrieval (`Div`, `TopK-Div`) yields more reliable in-context learning than similarity-only baselines (`TopK`).

**Tasks and datasets.** We consider 5 tasks: sentiment classification (classification task), commonsense reasoning (multiple-choice), text to SQL generation (generation), math (generation), and reading comprehension (generation). For sentiment classification, we test on SST-2 (Scarlatos and Lan, 2023), IMDB (Maas et al., 2011) and Amazon (polarity) (McAuley and Leskovec, 2013). For commonsense reasoning, we use ARC-Easy (Clark et al., 2018) and CommonsenseQA (CsQA) (Talmor et al., 2019). For text to SQL generation, we use GeoQuery (Zelle and Mooney, 1996; Tang and Mooney, 2001). For math problems, we test on GSM8K (Cobbe et al., 2021) and GSM-Plus-Mini (Li et al., 2024)

Table 1: (**Comparison of different in-context example selection methods**) We compare diversity-aware methods `Div` and `TopK-Div` with randomly chosen (`Rand`) and similarity-based method `TopK` on a variety of tasks using different models with different number of in-context examples $K$. For `TopK` and `TopK-Div`, we test ten different permutations of the demonstration due to the determined choice by these methods; For `Rand` and `Div`, we test ten different random seeds. We use the corresponding instruct-tuned model for math tasks (GSM8K and GSM-Plus-Mini) and base model for all other tasks. For TopK and TopK-Div methods - both being deterministic approaches - we computed outcomes across ten distinct example permutations. For Rand and Div methods, we report the averaged results across ten random seeds. There is a huge improvement when the shot number increases from 0 to 4 / 8, which demonstrates the effectiveness of our example selection. Due to the absence of prior knowledge for Geoquery in the zero-shot ($k = 0$) setting, we omit its $k = 0$ results. The bold entries indicate optimal performances. The `std` is no more than 1% in most cases; see Appendix D for details.

| Model | K | Method | Classification | | Multiple-choice | | GSM8K | Math GSM-Plus-Mini | Code GeoQuery | Reading | |
|---|---|---|---|---|---|---|---|---|---|---|---|
| | | | SST-2 | Amazon | ARC-Easy | CsQA | | | | SQuAD | SCIQ |
| Llama-3.1-8B | 0 | - | 87.50 | 95.40 | 82.43 | 62.80 | 53.45 | 65.12 | — | 42.30 | 36.40 |
| | 4 | Rand | 91.31 | 96.38 | 84.72 | 71.15 | 82.24 | 66.90 | 12.50 | 75.95 | 74.00 |
| | | TopK | **94.13** | 96.24 | **86.10** | 72.54 | 81.99 | 65.30 | 62.79 | 73.51 | 72.70 |
| | | Div-S3 | 92.89 | **96.81** | 85.81 | 72.28 | **82.52** | 66.97 | 34.54 | **77.95** | **74.61** |
| | | Div | 91.50 | 96.18 | 85.06 | 71.17 | 82.14 | **66.92** | 33.79 | 75.66 | 74.47 |
| | | TopK-Div | 92.75 | 96.43 | 85.83 | **72.57** | 81.74 | 66.12 | **71.14** | 73.28 | 73.87 |
| | 8 | Rand | 92.27 | 96.63 | 84.38 | 72.23 | 82.81 | 66.72 | 23.11 | 77.13 | 74.65 |
| | | TopK | 93.64 | 96.12 | **85.91** | **73.91** | 82.26 | 65.99 | 72.04 | 75.52 | 74.72 |
| | | Div-S3 | **93.65** | **96.74** | 85.50 | 73.04 | **83.00** | **66.82** | 43.61 | **79.41** | 75.15 |
| | | Div | 92.95 | 96.25 | 84.97 | 72.77 | 82.98 | 66.56 | 38.61 | 77.71 | **75.17** |
| | | TopK-Div | 93.33 | 96.57 | 85.39 | 73.76 | 82.63 | 66.48 | **78.68** | 76.13 | 75.07 |
| Gemma-2-9B | 0 | - | 67.50 | 85.10 | 88.15 | 61.80 | 16.07 | 32.79 | — | 37.90 | 41.10 |
| | 4 | Rand | 93.33 | 96.15 | 89.52 | 74.70 | 84.29 | 74.40 | 13.89 | 77.19 | 75.80 |
| | | TopK | **94.47** | 96.34 | **90.50** | 75.19 | 84.25 | 74.50 | 61.14 | 74.82 | 75.24 |
| | | Div-S3 | 93.64 | 96.54 | 89.98 | 75.51 | 84.07 | **74.86** | 37.32 | **77.94** | **76.13** |
| | | Div | 93.45 | 95.69 | 90.03 | 74.85 | **84.44** | 73.34 | 36.29 | 77.06 | 75.96 |
| | | TopK-Div | 93.34 | **96.57** | 90.19 | **75.60** | 83.54 | 74.47 | **70.43** | 75.05 | 75.21 |
| | 8 | Rand | 93.30 | 96.09 | 89.39 | 75.98 | **84.34** | 74.48 | 24.36 | 79.23 | 76.28 |
| | | TopK | **94.20** | 96.55 | **90.62** | 76.14 | 83.57 | 75.36 | 71.00 | 77.59 | 75.55 |
| | | Div-S3 | 93.38 | **96.60** | 90.10 | **76.98** | 83.56 | **75.62** | 45.86 | **79.79** | **77.24** |
| | | Div | 93.41 | 95.94 | 89.90 | 76.60 | 84.22 | 74.69 | 42.07 | 79.05 | 76.65 |
| | | TopK-Div | 94.04 | 96.58 | 90.48 | 76.53 | 83.85 | 75.16 | **76.32** | 77.64 | 76.24 |
| Mistral-7B-v0.3 | 0 | - | 66.50 | 94.00 | 76.41 | 51.80 | 9.48 | 5.17 | — | 30.50 | 34.20 |
| | 4 | Rand | 91.00 | 94.02 | 82.77 | 69.83 | 48.78 | 37.20 | 12.14 | **76.70** | 74.71 |
| | | TopK | **93.57** | **96.17** | **85.21** | 69.73 | 49.28 | 38.20 | 60.14 | 75.04 | 73.73 |
| | | Div-S3 | 92.83 | 95.60 | 83.93 | **70.29** | **51.43** | 38.22 | 37.75 | 75.74 | 75.54 |
| | | Div | 91.98 | 94.15 | 82.98 | 70.15 | 49.49 | 37.50 | 34.89 | 75.96 | **75.83** |
| | | TopK-Div | 92.73 | 95.90 | 84.55 | 69.91 | 49.99 | **38.45** | 71.46 | 74.43 | 73.16 |
| | 8 | Rand | 92.49 | 95.35 | 83.69 | 71.65 | 47.86 | 36.32 | 22.18 | 77.30 | 75.54 |
| | | TopK | **93.61** | **96.15** | **85.17** | 71.88 | 48.43 | 37.35 | 70.50 | 77.05 | 75.44 |
| | | Div-S3 | 92.79 | 96.10 | 84.38 | **72.47** | 48.57 | 36.60 | 45.86 | **77.71** | **76.56** |
| | | Div | 92.55 | 95.10 | 84.27 | 72.04 | 48.33 | 36.12 | 39.14 | 77.67 | 76.30 |
| | | TopK-Div | 93.47 | 96.11 | 84.85 | 71.81 | **48.60** | **37.81** | **77.93** | 77.44 | 75.22 |

datasets. For reading comprehension, we use SQuAD (Rajpurkar et al., 2016) and SCIQ (Welbl et al., 2017) datasets. We subsample some datasets to reduce the computation resources needed.

**Models.** Our main experiments are conducted on Llama 3.1 and Llama 3.2 (Dubey et al., 2024), Gemma 2 (Team et al., 2024), and Mistral v0.3 (Jiang et al., 2023) families of models. For math problems (GSM8K and GSM-Plus-Mini), we use the instruction-tuned LLMs, while for all other datasets, we use the base model. For the main experiments, we use Sentence-BERT (Reimers, 2019) to compute all the embeddings for `TopK`, `Div`, and `TopK-Div`. Experiments are conducted on 2 A100 GPUs.

**Hyperparameters.** For `Div`, we choose to first reduce the demonstration set $D$ to a "coreset" $D_r$ with size 100. This choice balances full similarity selection (`TopK`) and methods focusing mainly on diversity. For `TopK-Div`, we choose $\alpha = 1/2$ to balance between similarity and diversity. For `Div-S3`, we choose $|D_r| = 100$. For the classification task, we predict positive if the logit for token `great` is larger than that for token `terrible` for the next token prediction given a prompt. For multiple-choice tasks, we choose the option with the lowest average CE loss given a prompt. For generation tasks (text to SQL, math, reading comprehension), we use greedy decoding. More experimental details, including the prompt for each task, can be found in Appendix B.

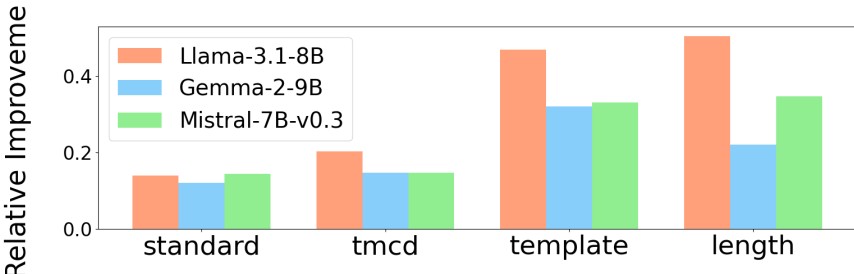

Figure 2: (**Comparison of different methods on GeoQuery OOD setting**) We report the relative improvement of `TopK-Div` over `TopK` when demonstrations and queries come from different GeoQuery dataset splitting ways. "standard" split denotes ID the setting. The relative improvement enlarges in the OOD setting.

## 3.1 MAIN FINDINGS

**Finding 1: Diversity-aware methods perform better on more "challenging" tasks.** Table 1 summarizes our main results in the in-distribution (ID) setting, where the demonstration distribution $\mathcal{D}_\mathcal{X}$ matches the query distribution $\mathcal{Q}_\mathcal{X}$. For simpler tasks like sentiment classification, `TopK` consistently performs best, significantly outperforming diversity-emphasizing methods like `Rand` and `Div` (e.g., `TopK` outperforms these methods by at least 1% on average in SST-2), while `TopK-Div` (balancing similarity and diversity) typically ranks between these extremes. In commonsense reasoning (multiple-choice), introducing some diversity via `TopK-Div` improves performance over pure `TopK`, as observed in Commonsense QA, although the gains on ARC-Easy remain modest.

For more complex tasks—including reading comprehension, text-to-SQL generation, multi-step mathematical reasoning, and GeoQuery—introducing diversity consistently improves performance over the similarity-based `TopK` baseline. In GeoQuery specifically, diversity (`TopK-Div`) yields at least a 7% absolute accuracy gain, likely due to enhanced feature coverage (Levy et al., 2023; Ye et al., 2023).[1] However, excessive diversity (`Rand` and `Div`) becomes detrimental, as overly dissimilar examples fail to illustrate coherent solution patterns.

For math and reading comprehension tasks, methods that emphasize diversity—such as `Div` and even `Rand` —outperform `TopK`. Interestingly, the effectiveness of random selection cannot be explained solely by coverage, as random demonstrations do not systematically capture similar problem structures. Instead, we observe that for tasks where the model already exhibits strong zero-shot abilities (e.g., Math and Reading), incorporating diverse demonstrations encourages the model to rely more on its general reasoning skills rather than memorizing surface-level patterns. To support this interpretation, we present 0-shot and 1-shot performance results in Appendix C.1, which highlight the model's underlying capabilities.

Regarding `Div-S3`, we observe that it underperforms `TopK` on relatively simple classification tasks, but shows a performance advantage on more complex tasks such as Math and Reading. These trends are consistent with other diversity-based methods and complement the analysis in (Kumari et al., 2024), extending their findings to a broader range of task types and difficulty levels.

To further verify the impact of diversity on different tasks, we also conducted experiments on both `Div` and `TopK-Div` by varying their degree of diversity (Figure 4 and Table 7). We observe a consistent pattern across both `Div` and `TopK`: for simpler tasks, introducing less diversity (i.e., employing larger subset size $m$ for `Div` or higher $\alpha$ for `TopK-Div`) leads to better performance, whereas for more complex tasks, incorporating greater diversity yields superior performance. Due to space limit, we defer a more detailed discussion to Appendices C.2 and C.3.

**Finding 2: Diversity helps out-of-distribution generalization.** Table 2 presents results on sentiment classification, commonsense reasoning, and reading comprehension, while Figure 2 shows text-to-SQL generation performance in the out-of-distribution (OOD) setting, where the demonstration distribution $\mathcal{D}_\mathcal{X}$ and query distribution $\mathcal{Q}_\mathcal{X}$ differ.

---

[1]The correspondence between inputs and outputs is deterministic, and the model need to learn this mapping from the provided examples. Better coverage of the query inputs implies that the model acquires a larger portion of the mappings required for the query inputs.

Table 2: **(Comparison of different methods when demonstration and query come from different distribution)** We compare the methods on different tasks. The number of shots is fixed as $K = 4$. We observe that diversity-aware methods are more robust to out-of-distribution query. The performance drop from ID to OOD on `TopK` is in general larger than diversity-aware methods.

| | Test | Demo. | Rand | TopK | Div-S3 | Div | TopK-Div |
|---|---|---|---|---|---|---|---|
| Llama-3.1-8B | SST-2 | SST-2 | 91.31 | **94.13** | 92.89 | 91.50 | 92.75 |
| | | IMDB | 88.85 | **90.80** | 86.90 | 90.71 | **90.80** |
| | | Amazon | 88.28 | 89.50 | **90.70** | 86.64 | 89.60 |
| | CsQA | CsQA | 71.15 | 72.54 | 72.28 | 71.17 | **72.57** |
| | | ARC-Easy | 66.86 | 66.70 | 66.50 | 67.08 | **67.70** |
| | SCIQ | SCIQ | 74.00 | 72.70 | **74.61** | 74.47 | 73.87 |
| | | SQuAD | 72.11 | 71.40 | **73.67** | 72.79 | 71.60 |
| Gemma-2-9B | SST-2 | SST-2 | 93.33 | **94.47** | 93.64 | 93.45 | 93.34 |
| | | IMDB | 88.66 | 89.90 | 85.50 | 88.59 | **91.10** |
| | | Amazon | 88.69 | 89.40 | 89.30 | **90.49** | 89.60 |
| | CsQA | CsQA | 74.70 | 75.19 | 75.51 | 74.85 | **75.60** |
| | | ARC-Easy | 68.30 | 68.90 | 68.80 | 68.58 | **69.50** |
| | SCIQ | SCIQ | 75.80 | 75.24 | **76.13** | 75.96 | 75.21 |
| | | SQuAD | 73.63 | 73.60 | **76.07** | 74.64 | 73.50 |

Table 3: Relative improvement of `TopK-Div` over `TopK` on GeoQuery and SQuAD on different sets of the queries. For the GeoQuery dataset, we fine-tuned both base models on its training set. We categorized questions in testing set as "Easy" if the fine-tuned models correctly answered them in a 0-shot setting, and as "Hard" if these models failed to answer them correctly in the same 0-shot setting. We report the performance of both methods in a 4-shot setting. For SQuAD, we split the testing set only using the fine-tuned gemma-2-9B model, since fine-tuning the Llama-3.1-8B model yielded poor results. We observe that `TopK-Div` exhibits greater improvement on "Hard" examples.

| Split | Method | Gemma-2-9B | | Llama-3.1-8B | |
|---|---|---|---|---|---|
| | | GeoQuery | SQuAD | GeoQuery | SQuAD |
| Easy | TopK | 72.09 | 83.01 | 79.31 | 81.04 |
| | TopK-Div | 77.91 | 82.66 | 83.71 | 79.65 |
| Hard | TopK | 56.29 | 20.00 | 51.52 | 24.44 |
| | TopK-Div | 67.11 | 23.70 | 62.13 | 25.19 |

Overall, diversity improves OOD in-context learning. In sentiment classification, `TopK` performs best when both demonstrations and queries come from SST-2. However, when demonstrations shift to IMDB (another movie review dataset), `TopK` and `TopK-Div` perform similarly. When using Amazon (a shopping review dataset) as demonstrations, `TopK-Div` surpasses `TopK`. A similar trend is observed in commonsense reasoning: replacing Commonsense QA (ID) demonstrations with ARC-Easy (OOD) increases the performance gap between `Div` and `TopK` from 0.4% to 1.0%. Text-to-SQL generation follows this pattern, with a larger improvement in OOD settings. Additionally, we note that GSM-Plus-Mini serves as an OOD setting for GSM8K (Math in Table 1), as they share the same training set. A larger improvement from adding diversity is also observed on GSM-Plus-Mini.

For reading comprehension, switching to an OOD demonstration dataset does not significantly widen the gap between `Div` and `TopK`, but `Div` still outperforms `TopK`. We provide additional out-of-distribution (OOD) results in Appendix C.4, which further reinforce our conclusions.

Beyond explicitly defined OOD settings, the contrast between the Amazon and SST-2 classification tasks in Table 1 further illustrates the impact of distributional differences on the effectiveness of diversity-based selection. While SST-2 consists of curated movie reviews with relatively homogeneous content—where `TopK` consistently outperforms diversity-based methods—Amazon reviews span heterogeneous domains such as electronics, books, and household items. This broader domain variability in Amazon leads to performance gains for diversity-driven methods like `Rand`, `Div-S3` and `Div`, sometimes even surpassing `TopK`. These results are consistent with the patterns observed in Table 2 and provide additional empirical support for our Finding 2.

**Finding 3: Diversity performs better on harder examples.** Besides discussing the performance of diversity-aware methods (`TopK-Div`, `Div`, and even `Rand`) at task levels, we also analyze which specific examples benefit most from diversity. For this, we first provide a method to quantify the "difficulty level" of examples. Motivated by (Swayamdipta et al., 2020), we use whether a model can correctly answer a question after fine-tuning as an indicator of that question's difficulty for a specific language model. Therefore, we fine-tuned the corresponding base model on the dataset's training set using LoRA. Subsequently, based on whether this fine-tuned model could accurately answer questions in the testing set under a zero-shot setting, we classified these questions as "easy" or "hard".

We examine this phenomenon in GEOQUERY and SQUAD, where `TopK-Div` consistently outperforms `TopK`. Table 3 shows that diversity yields greater benefits on harder examples. In GEOQUERY, the absolute accuracy improvement of `TopK-Div` over `TopK` is 5.11% on easy examples (averaged across two models), increasing to 10.72% on hard examples. In SQUAD, while `TopK-Div` slightly underperforms `TopK` on easy examples by 0.87%, it outperforms `TopK` on hard examples by 2.23%.

## 3.2 UNDERSTANDING THE ROLE OF DIVERSITY: BEYOND COVERAGE EFFECTS

We examine how diversity contributes to in-context learning (ICL) performance, distinguishing between its impact through *coverage* and through *mechanisms beyond coverage*, at both the example level and the task level.

**Example-level analysis.** In the GeoQuery dataset, the observation that diversity performs better on harder examples (see Table 3) aligns with the notion of enhanced coverage: difficult examples often require modeling more nuanced or rare local structures (Levy et al., 2023; Gupta et al., 2023), which diversity-based methods are more likely to capture.

However, in SQuAD, we observe a different pattern. Even when $k = 1$, `TopK` underperforms compared to `Rand` and `Div`, suggesting that coverage alone is insufficient to explain the performance gap. To probe this, we remove irrelevant noisy examples from the SQuAD dataset and rerun the comparison. This cleaning significantly improves `TopK` and `TopK-Div`, but has minimal impact on `Rand` and `Div` —indicating that the strength of diversity-based methods extends beyond simple structural alignment with the query.

We present detailed results and analysis in Appendix C.5 to support this claim.

**Task-level analysis.** As shown in Table 7, increasing the number of demonstrations (e.g., from 4 to 8 shots) magnifies the benefits of diversity. While coverage-based reasoning suggests this may be due to broader inclusion of features, our findings point to a richer effect.

Specifically, when given more demonstrations, models appear to better synthesize the overall conceptual structure of the task. Diversity enables this by exposing the model to varied facets of the task distribution, which helps form a more general and transferable representation. In contrast, with fewer shots, the model has limited capacity to form such abstractions, and similarity alone may suffice.

We further justify this effect theoretically in Appendix E, showing that diversity supports a form of generalization that cannot be fully explained by coverage alone.

## 3.3 PRACTICAL INSIGHTS

In summary, our findings provide the following guidelines for selecting demonstrations in ICL:

1. **Leverage similarity for simple tasks.** When the task is relatively easy (e.g., sentiment classification) and the model already exhibits sufficient ability, selecting demonstrations purely based on similarity to the query is generally sufficient to elicit strong performance.

2. **Use diversity to bridge distribution gaps.** When there is a significant mismatch between the test distribution and the available demonstration pool, incorporating diversity in selection helps the model generalize better by exposing it to a broader range of examples.

3. **Favor diversity for complex or knowledge-intensive tasks.** For tasks that require the model to extract and apply task-solving knowledge (e.g., math reasoning or reading comprehension), selecting diverse demonstrations provides broader coverage of relevant patterns or skills.

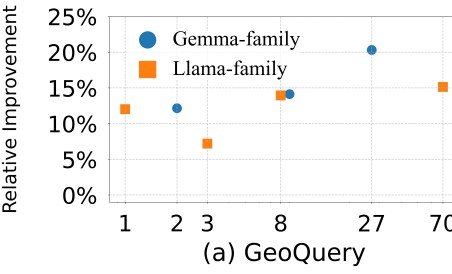 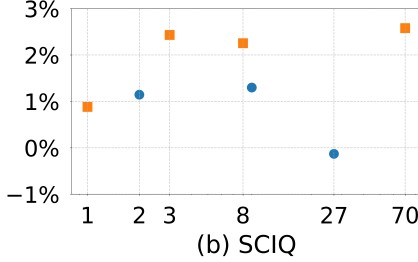

Figure 3: The relative improvement of diversity-aware methods over `TopK`. **Left:** relative improvement of `TopK-Div` over `TopK` on GeoQuery standard split. **Right:** relative improvement of `Div` over `TopK` on SCIQ.

4. **Adapt to noise levels.** For *low-noise datasets*, coverage-oriented (similarity-based) selection is more effective, as it aligns demonstrations closely with the query and helps the model lock onto the correct input-output mapping. In contrast, for *high-noise datasets*, increased diversity is beneficial to reduce overfitting to spurious correlations and enhance robustness.

### 3.4 ABLATION STUDIES

We conduct experiments to observe how the improvement of diversity-aware methods over `TopK` changes if we change the size of the LLM used, since it is possible that, as the models scale up, ICL is not sensitive to data selection, and thus the improvement of diversity over pure similarity diminishes.

Figure 3 shows the relative improvement on GeoQuery standard split and SCIQ, where we observe the clear benefit of the diversity-aware method in Section 3.1, on Llama-3.1/3.2 and Gemma-2 families with different sizes. We observe that in these two tasks, in general, the relative improvement does not decrease that much even if the model scales up, which indicates the importance of understanding the role of diversity in demonstration selection.

In Appendix D, we present experiments on additional models. We also report results across a range of settings, including fine-grained variations in $k$, different subset sizes for the `Div` method, fixed training sets, and changes in the embedding or decoding strategies. In particular, we implement a **purely** diversity-based method, `K-Means`, whose diversity score can exceed that of `Div`. Its superior performance on Math and Reading tasks further supports Finding 1. These ablation studies consistently reinforce our main findings, demonstrating the generality and robustness of our conclusions.

## 4 CONCLUSION, LIMITATIONS, AND FUTURE WORKS

We investigate the role of diversity in retrieval-based demonstration selection for in-context learning (ICL). Across a wide range of tasks and multiple model families, we find that incorporating diversity into selection strategies consistently improves performance, especially when the task is difficult, the query is challenging, or there is a distribution shift between the query and available demonstrations. These findings are further supported by comprehensive ablation studies.

In addition, we provide theoretical justification that explains when and why diversity offers advantages over purely similarity-based selection. Together, our empirical and theoretical insights offer practical guidance for selecting effective demonstrations in ICL and deepen the understanding of diversity's role in prompting large language models.

Note that the internal mechanism behind why diversity benefits still remains unclear. Part of our findings can be explained by coverage, which is aligned with previous literature, but the superior performance on math, reading comprehension, and OOD generalization, cannot be explained by simply incentivizing coverage. Potential future research directions include both theoretical and empirical explorations into why diversity aids demonstration selection beyond coverage. This could involve deeper analysis of model representations, interactions between diverse demonstrations, or alternative explanations grounded in information theory or representation learning. Additionally, our diversity heuristic is tested on English text only; cross-lingual robustness is left for future work.

## REPRODUCIBILITY STATEMENT

We have included in the supplementary materials the complete codebase used in our ICL experiments, with all programs fully anonymized. In addition, the data folder contains all processed datasets employed in our study. We guarantee that running the provided code will reproduce the results reported in the paper.

## ETHICS STATEMENT

This research fully aligns with the ethical principles outlined in the ICLR Code of Ethics, especially in its commitment to responsible stewardship of AI research. Our work systematically investigates the role of sample-selection diversity in *In-context learning (ICL)*, aiming to improve the performance, robustness, and reliability of large language models.

The primary motivation is to contribute positively to society and human well-being. By demonstrating that diversity-aware selection of in-context examples can lead to improvements on complex tasks (such as mathematical reasoning and code generation) and out-of-distribution queries, we hope to foster AI systems that generalize better, thereby serving societal applications in research, education, software development, and beyond. We pay particular attention to underrepresented or challenging settings, in line with the ICLR principle of giving emphasis to less-advantaged groups.

In striving for scientific excellence, we adhere to methodological rigor, transparency, and reproducibility. Our conclusions are supported by systematic experiments across multiple tasks, datasets, and models (including LLaMA 3.1, Gemma 2, Mistral-v0.3), and by a theoretical framework that elucidates why diversity helps. We use publicly available benchmark datasets (e.g. SST-2, GSM8K, GeoQuery) and open-source models, and we provide full details of experimental design, hyperparameters, and evaluation procedures in the paper and appendix.

We also commit to fairness and non-discrimination. Although bias mitigation is not a direct focus, our findings suggest that diversity-aware in-context selection can improve out-of-distribution robustness, potentially helping models maintain performance even for underrepresented groups or non-mainstream distributions. We view this as a positive step toward more equitable AI systems.

Privacy and respect for intellectual labor are also core commitments. Our study uses only publicly available, anonymized datasets; no new personal or sensitive data were collected, and no human subjects were involved. We fully cite all utilized datasets, models, and prior work, giving due credit to others' contributions.

In summary, we believe this work is a responsible and beneficial contribution toward building more robust and trustworthy large language models. We have carefully considered the relevant ethical dimensions and commit to conducting our research according to the scientific and ethical standards expected under ICLR's Code of Ethics.

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

CONTENTS

## A  MORE RELATED WORKS

**Demonstration selection**  Retrieval-based demonstration selection for ICL has long been studied, and the most notable methods are the *similarity*-based methods (Liu et al., 2021; Yang et al., 2022; Wu et al., 2023; Qin et al., 2023). These are often augmented by trainable deep learning retrievers aimed at capturing core skills or features beyond mere semantic similarity (Karpukhin et al., 2020; Rubin et al., 2022; Luo et al., 2023; Scarlatos and Lan, 2023; An et al., 2023b), or by incorporating LLM feedback for refinement (Li and Qiu, 2023a; Chen et al., 2023; Wang et al., 2023). Conversely, diversity-based, or more accurately, coverage-based methods are less prevalent in retrieval-based selection. Existing studies in this vein typically address tasks with clear local structures where feature coverage is advantageous (Levy et al., 2023; Ye et al., 2023; Gupta et al., 2023; An et al., 2023a). For non-retrieval-based ICL, where a fixed set of demonstrations is selected for a specific task, diversity is recognized as beneficial (Zhang et al., 2023b; Gao et al., 2023; Su et al., 2023; Yang et al., 2023).

**Understanding in-context learning**  Efforts to understand ICL span both theoretical and empirical investigations. Theoretical perspectives often frame ICL as either a Bayesian inference procedure (Xie et al., 2022; Wang et al.; Wies et al., 2023; Jiang, 2023; Zhang et al., 2023a) or an implicit form of meta-optimization akin to gradient descent (Dai et al., 2023; Von Oswald et al., 2023a;b; Deutch et al., 2024; Shen et al., 2023). Research on ICL for regression tasks (Garg et al., 2022; Li et al., 2023b;a; Akyürek et al., 2023) provides valuable insights; notably, (Akyürek et al., 2023) suggest transformers can identify min-norm solutions in-context for linear regression, a finding that supports the role of demonstration diversity. Empirical studies have further examined factors such as input-label mapping (Min et al., 2022; Yoo et al., 2022; Pan et al., 2023), the influence of demonstration order (Lu et al., 2022; Liu et al., 2024), and the importance of calibration for ICL efficacy (Zhao et al., 2021).

## B  MORE EXPERIMENT DETAILS

### B.1  PROMPT TEMPLATE

Table 4 lists the template we use for different tasks. We take $K = 2$ as an example.

### B.2  DATASET DETAILS

To reduce computational cost, we performed random sampling on both the *demo* and *test* set for classification, multiple-choice and reading tasks. For classification tasks, the sampled datasets from IMDB and SST-2 are consistent with Chang and Jia (2023). A fixed random seed of 42 was used for all sampling procedures. For math tasks, since the *test* set sizes of PRM800K and GSM8K datasets are close to the sampled *test* set sizes of other tasks, we directly used their existing *demo* and *test* set. Detailed sampling statistics are provided in Table 5.

### B.3  EVALUATION DETAILS

For the sentiment classification task (classification), given the prompt listed in Table 4, we compute the logit for "great" and "terrible" respectively, and predict the sentiment to be positive if the logit for "great" is larger than that for "terrible", and vice versa. We report the accuracy metric.

For commonsense reasoning tasks (multiple-choice), given the prompt, we compute the average cross-entropy loss on each given option, conditioned on the prompt. Then we pick the option with the smallest average cross-entropy loss. We report the accuracy metric.

For reading comprehension (generation), given the prompt, we generate the answer using greedy decoding. We stop if we generate one of the following string: "\n\n", "\n\n\n", "Support", "Support:", "Question", "Question:". We compare the generated answer with the gold answer, and report the exact match metric. There are several optional answers for the squad test sample, if the generated answer exactly matches one of them, we consider it correct.

Table 4: Prompt template for different tasks with 2 demonstrations. For Math problems, we also apply the chat template since we use the instruct models (done by applying the function "apply_chat_template" on the instruct models' tokenizer).

| Name | Template |
|---|---|
| Sentiment Classification (SST-2, IMDB, Amazon) | Question: {input_1}
Answer: {output_1}

Question: {input_2}
Answer: {output_2}

Question: {input_query}
Answer: |
| Commonsense Reasoning (ARC-Easy, CsQA) | Question: {input_1}
Answer: {output_1}

Question: {input_2}
Answer: {output_2}

Question: {input_query}
Answer: |
| Reading Comprehension (SQuAD, SCIQ) | Support: {support_1}
Question: {input_1}
Answer: {output_1}

Support: {support_2}
Question: {input_2}
Answer: {output_2}

Support: {support_query}
Question: {input_query}
Answer: |
| text to SQL (Geo-Query) | Question: {input_1}
Answer: {output_1}

Question: {input_2}
Answer: {output_2}

Question: {input_query}
Answer: |
| Math (GSM8K, PRM800K) | Question: {input_1}
Answer: {output_1}

Question: {input_2}
Answer: {output_2}

Let's think step by step. You need to solve the final
↪ question and answer in the format: \n#### \{result\}
Question: {input_query}
Answer: |

Table 5: **Detailed dataset size before and after sampling.** We show the original and sampled size of demonstration set and test set for all dataset we considered.

| Dataset size | Classification | | | Multiple-choice | | | Math | | | Code | Reading | |
| | SST-2 | Amazon | Imdb | ARC-Easy | CsQA | PRM800K | GSM8K | GSM-Plus-Mini | | GeoQuery | SQuAD | SCIQ |
| --- | --- | --- | --- | --- | --- | --- | --- | --- | --- | --- | --- | --- |
| Sampled demo set | 1000 | 1000 | 1000 | 1000 | 1000 | 12000 | 7473 | 7473 | | 600 | 10000 | 1000 |
| Sampled test set | 1000 | 1000 | 1000 | 1000 | 1000 | 500 | 1319 | 2400 | | 280 | 1000 | 1000 |
| Original demo set | 67300 | 3600000 | 25000 | 2250 | 9740 | 12000 | 7473 | 7473 | | 600 | 87600 | 11700 |
| Original test set | 1820 | 400000 | 25000 | 2380 | 1140 | 500 | 1319 | 2400 | | 280 | 10600 | 1000 |

Table 6: Performance of 0-shot and 1-shot Baseline in Code and Reading Tasks. When $k = 1$, there is only one possible permutation, so we report a single result for both TopK and TopK-Div methods. For Rand and Div approaches, we report the averaged results across ten random seeds. Embedding = all-roberta-large-v1.

| Model | Dataset | $K = 0$ | $K = 1$ | | | | $K = 4$ | | | |
| | | - | Rand | Topk | Div | Topk-Div | Rand | Topk | Div | Topk-Div |
| --- | --- | --- | --- | --- | --- | --- | --- | --- | --- | --- |
| Llama-3.1-8B | Code (Geoquery) | − | 2.61 | 37.14 | 16.93 | 37.14 | 12.57 | 63.04 | 33.71 | 71.07 |
| | Reading (SQuAD) | 42.30 | 68.64 | 67.00 | 67.87 | 67.00 | 75.95 | 73.51 | 75.66 | 73.28 |
| Gemma-2-9B | Code (Geoquery) | − | 3.07 | 41.43 | 16.71 | 41.43 | 13.89 | 61.14 | 36.29 | 70.43 |
| | Reading (SQuAD) | 37.90 | 71.34 | 69.00 | 70.69 | 69.00 | 77.19 | 74.82 | 77.06 | 75.05 |
| Mistral-7B-v0.3 | Code (Geoquery) | − | 2.75 | 40.71 | 18.39 | 40.71 | 12.14 | 60.14 | 34.89 | 71.46 |
| | Reading (SQuAD) | 30.50 | 69.12 | 66.30 | 67.80 | 66.30 | 76.70 | 75.04 | 75.96 | 74.43 |

For text to SQL (generation), given the prompt, we generate the answer using greedy decoding. We stop if we generate one of the following string: "\n\n", "\n\n\n", "Question", "Question:". We compare the generated answer with the gold answer, and report the exact match metric.

For math problem (generation), given the prompt, we generate the answer using greedy decoding. We do not stop the generation process unless the instruct model generates the stop sign itself. We first try to extract the math expression from the following format "#### {expression}". If failed, we try to extract from the following format "\{boxed}{expression}". If both failed, we extract the final math expression from the answer. The report exact match metric.

For each task, the selected examples in TopK and TopK-Div are fixed, and these two methods are tested once. For Rand and Div, where example selection involves randomness, we test with ten random seeds and report the average results.

## C    ADDITIONAL EXPERIMENTS

In this section, we present some addition (supplementary) experiment results for Section 3. This section is structured as follows:

- Appendix C.1 shows the results of different tasks under 0-shot or 1-shot, to justify the effectiveness of in-context examples;

- Appendix C.2 discusses the best subset_size in `Div`;

- Appendix C.3 illustrates the gap between different levels of diversity in `TopK-Div`;

- Appendix C.4 includes more results and discussions for the OOD setting;

- Appendix C.5 contains the detailed experiments that imply the effect of diversity that is beyond coverage.

### C.1    RESULTS OF 0/1-SHOT

To verify whether the model inherently possesses the ability to solve certain tasks, we tested its 0-shot and 1-shot performance on the SQuAD and GeoQuery datasets. For the Reading task, accuracy is calculated only when the output exactly matches the answer, imposing strict format requirements. Consequently, on SQuAD, once the model understood the output format in the 1-shot setting, the absolute performance gap compared to the 4-shot setting was less than 8%. However, on GeoQuery,

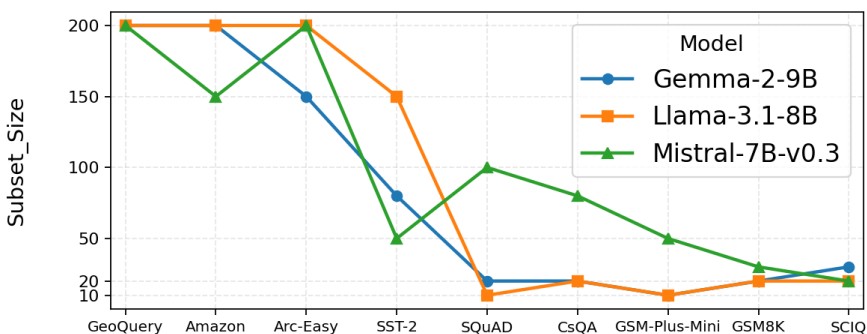

Figure 4: **(Optimal Subset_Size of the `Div` Method for Different Tasks)** We report the optimal subset_size of the `Div` method across different tasks. The results show that relatively easier tasks, such as Classification and Multiple-choice, tend to favor larger subset_size values (with data points concentrated on the left side of the x-axis), whereas more challenging tasks, such as Math and Reading, exhibit substantially smaller optimal subset_size values, with an overall average not exceeding 30 (with data points concentrated on the right side of the x-axis).

even after the model grasped the output format via the 1-shot example, the absolute performance gap compared to the 4-shot setting was still over 20%.

Therefore, the model possesses a strong inherent ability to solve the Reading task (a similar conclusion also holds for the Math task). Conversely, the model itself lacks domain-specific knowledge related to GeoQuery and thus needs to learn more from the provided context.

## C.2 ADDITIONAL SUPPLEMENT OF FINDING 1: ADJUSTING SUBSET SIZES FOR DIV

For each task, we identify the subset_size that yields the best performance for `Div` ($subset\_size \in \{10, 20, 30, 50, 80, 100, 120, 150, 200\}$). On simpler tasks (e.g., Classification and Multiple-choice), the average optimal subset_size exceeds 100 (Figure 4), indicating that introducing less diversity is more beneficial. In contrast, on more complex tasks (e.g., Math and Reading), the average optimal subset_size is below 30, suggesting that incorporating more diversity is advantageous . GeoQuery serves as a typical example of a task that requires strong coverage, for which `Div` with small subset_size fails to achieve satisfactory performance.

## C.3 ADDITIONAL SUPPLEMENT OF FINDING 1: ADJUSTING $\alpha$ FOR TOPK-DIV

Let $\mathrm{Acc}_\alpha$ denotes the accuracy of `TopK-Div` parameterized by $\alpha$ (Equation (3)), We define:

$$\Delta = \tfrac{1}{5} \sum_{i=6}^{10} \mathrm{Acc}_{i/10} - \tfrac{1}{5} \sum_{i=1}^{5} \mathrm{Acc}_{i/10}. \tag{4}$$

The difference $\Delta$ quantifies the gap between the average accuracy of lower diversity and higher diversity in `TopK-Div` (e.g, $\Delta > 0$ means less diversity is better). As shown in Table 7, minimal diversity is optimal for simpler tasks, while higher diversity consistently enhances performance as task difficulty increases.

## C.4 MORE RESULTS ON OOD SETTING

In this part, we show the OOD results of math problems on Llama-3.1-8B/70B and Gemma-2-9B/27B instruct-tuned models. We use GSM8K as the demonstration set and PRM800K (Lightman et al., 2023) as the query set. Table 8 summarizes our result. We observe that diversity-aware methods are more robust to this distribution shift. Even for Gemma models where `TopK` performs very well on ID tasks (demonstration and query set are all PRM800K), `TopK` is outperformed by diversity-aware methods on the OOD setting. A similar trend also holds for Llama models. One interesting finding is that for PRM800K, more demonstration might not lead to better performance, and also in our experiment, using GSM8K as demonstration works better than using PRM800K data as demonstrations.

Table 7: **Comparison of `TopK-Div` results with different** $\alpha$. We report $\Delta$ (Equation (4)) for Classification, Multiple-choice and Reading task across six datasets. For each value of $\alpha$ in `TopK-Div`, we tested ten permutations and calculated the mean. For relatively simple tasks (SST-2, Amazon, ARC-Easy and CsQA), the average value 0.27% of $\Delta$ indicates that incorporating less diversity is more beneficial. In contrast, for relatively complex tasks (SCIQ, SQuAD), the average value -0.44% of $\Delta$ suggests that incorporating more diversity is advantageous. Specifically, the average $\Delta$ for SST-2 is 0.42%, for ARC-Easy is 0.34%, for CsQA is 0.07%, and for SCIQ is -0.53%. This trend is consistent with our understanding of task difficulty.

| Model | $K$ | $\Delta$ | | | | | |
|---|---|---|---|---|---|---|---|
| | | SST-2 | Amazon | ARC-Easy | CsQA | SCIQ | SQuAD |
| Llama-3.2-3B | 4 | 0.11% | 0.12% | 0.45% | 0.41% | -0.80% | 0.40% |
| | 8 | 1.05% | -0.01% | 0.20% | 0.06% | -0.74% | -1.01% |
| Gemma-2-2B | 4 | 0.34% | 0.50% | 0.87% | -0.37% | -0.07% | 0.04% |
| | 8 | 0.19% | 0.27% | -0.15% | 0.18% | -0.49% | -0.82% |

Table 8: **(Comparison of different methods on math when demonstration and query come from different distribution)** OOD setting for math problem where the test dataset is chosen to be PRM800K. We find that, diversity-aware methods are more superior than `TopK` in OOD setting. The method that achieves the best in each setting is highlighted.

| Model | Shots | Demo. | Rand | TopK | Div | TopK-Div |
|---|---|---|---|---|---|---|
| Llama-3.1-8B | $K = 4$ | PRM800K | 43.50 | 41.40 | **44.86** | 44.80 |
| | | GSM8K | 41.50 | 41.00 | **43.28** | 42.00 |
| | $K = 8$ | PRM800K | 43.32 | 43.00 | **44.28** | 40.00 |
| | | GSM8K | 41.66 | 42.00 | **43.46** | 40.80 |
| Llama-3.1-70B | $K = 4$ | PRM800K | 57.78 | 58.20 | 57.42 | **59.40** |
| | | GSM8K | 60.62 | **62.00** | 61.88 | 61.00 |
| | $K = 8$ | PRM800K | 54.72 | **59.00** | 55.86 | 58.00 |
| | | GSM8K | **61.14** | 59.60 | 60.96 | 60.00 |
| Gemma-2-9B | $K = 4$ | PRM800K | 38.04 | 42.40 | 36.78 | **44.40** |
| | | GSM8K | **42.10** | 41.00 | 41.04 | 42.20 |
| | $K = 8$ | PRM800K | 40.66 | **46.20** | 39.20 | 44.40 |
| | | GSM8K | 42.06 | 41.80 | **42.74** | 41.60 |
| Gemma-2-27B | $K = 4$ | PRM800K | 46.06 | 49.20 | 47.80 | **49.60** |
| | | GSM8K | 46.06 | 46.00 | **46.30** | 45.20 |
| | $K = 8$ | PRM800K | 47.10 | **50.40** | 47.04 | 48.60 |
| | | GSM8K | 45.40 | 44.80 | **45.92** | 45.20 |

## C.5 RESULTS ON PERTURBATION OF DATASETS

To explore whether the way diversity works is by achieving better coverage, we noticed that even when $k = 1$, `TopK` still underperforms `Rand`/ `Div` methods. We speculate this is because the support in the original dataset contains a lot of noise, causing similar examples not only to fail to provide effective information but also potentially to mislead the model into focusing on noisy information ("coverage" isn't helpful in such case).

Using DeepSeek-R1, we removed information irrelevant to the answer from the support passages in SQuAD, reducing content by approximately 50%. Based on this, we constructed two variants: SQuAD-Cut, where only the training set is streamlined, and SQuAD-Both-Cut, where both the training and test sets are streamlined. As shown in Table 9, the more streamlined (i.e., higher-quality and less noisy) the dataset, the better the performance of `TopK` and `TopK-Div`. Notably, their improvement margins are significantly larger than that of `Div` (though still more than 1% lower than `Div`). This indicates that when the dataset quality is higher, the "Coverage" mechanism can focus on

Table 9: Results for SQuAD with cut perturbation. We performed content trimming on the support portion of the SQuAD dataset using Deepseek-r1, retaining only the top 1/3 most answer-relevant content. SQuAD-Cut refers to trimming applied solely to the testing set, while SQuAD-Both-Cut indicates trimming performed on both testing and training sets. The values in parentheses represent performance improvements relative to the original SQuAD dataset.

| Model | $K$ | Dataset | Method | | | |
| | | | Rand | Topk | Div | Topk-Div |
|---|---|---|---|---|---|---|
| Llama-3.1-8B | 1 | SQuAD | 68.64 | 67.00 | 67.87 | 67.00 |
| | | SQuAD-Cut | 69.43 (+0.79) | **68.20 (+1.20)** | 68.45 (+0.58) | 67.70 (+0.70) |
| | | SQuAD-Both-Cut | 69.71 (+1.07) | **69.90 (+2.90)** | 69.47 (+1.60) | **69.90 (+2.90)** |
| | 4 | SQuAD | 75.95 | 73.51 | 75.66 | 73.28 |
| | | SQuAD-Cut | 77.15 (+1.2) | 75.96 (+2.45) | 77.00 (+1.34) | **76.89 (+2.61)** |
| | | SQuAD-Both-Cut | 76.95 (+1.00) | 76.15 (+2.64) | 77.76 (+2.10) | **76.47 (+3.19)** |
| | 8 | SQuAD | 77.13 | 75.52 | 77.71 | 76.13 |
| | | SQuAD-Cut | 79.10 (+1.97) | 77.43 (+1.91) | **79.43 (+1.72)** | 78.64 (+2.51) |
| | | SQuAD-Both-Cut | 79.39 (+2.26) | **78.66 (+3.14)** | 79.26 (+1.55) | 79.13 (+3.00) |
| Gemma-2-9B | 1 | SQuAD | 71.34 | 69.00 | 70.69 | 69.00 |
| | | SQuAD-Cut | 72.96 (+1.62) | **71.20 (+2.20)** | 72.25 (+1.56) | 71.10 (+2.10) |
| | | SQuAD-Both-Cut | 73.14 (+1.80) | **72.30 (+3.30)** | 72.67 (+1.98) | **72.30 (+3.30)** |
| | 4 | SQuAD | 77.19 | 74.82 | 77.06 | 75.05 |
| | | SQuAD-Cut | 78.75 (+1.56) | 77.64 (+2.82) | 78.23 (+1.17) | **78.65 (+3.60)** |
| | | SQuAD-Both-Cut | 78.72 (+1.53) | **77.47 (+2.65)** | 78.54 (+1.48) | 76.78 (+1.73) |
| | 8 | SQuAD | 79.23 | 77.59 | 79.05 | 77.64 |
| | | SQuAD-Cut | 80.41 (+1.18) | 79.74 (+2.15) | 80.45 (+1.40) | **80.72 (+3.08)** |
| | | SQuAD-Both-Cut | 80.22 (+0.99) | **79.05 (+1.46)** | 80.10 (+1.05) | 78.84 (+1.20) |
| Mistral-7B-v0.3 | 1 | SQuAD | 69.12 | 66.30 | 67.80 | 66.30 |
| | | SQuAD-Cut | 71.38 (+2.26) | **69.70 (+3.40)** | 69.21 (+1.41) | **69.70 (+3.40)** |
| | | SQuAD-Both-Cut | 71.44 (+2.32) | **70.70 (+4.40)** | 72.00 (+4.20) | **70.70 (+4.40)** |
| | 4 | SQuAD | 76.70 | 75.04 | 75.96 | 74.43 |
| | | SQuAD-Cut | 77.78 (+1.08) | 77.78 (+2.74) | 77.18 (+1.22) | **78.70 (+4.37)** |
| | | SQuAD-Both-Cut | 77.76 (+1.06) | 77.92 (+2.88) | 77.89 (+1.93) | **77.40 (+2.97)** |
| | 8 | SQuAD | 77.30 | 77.05 | 77.67 | 77.44 |
| | | SQuAD-Cut-R1 | 79.00 (+1.70) | **79.09 (+2.04)** | 78.64 (+0.97) | 79.07 (+1.63) |
| | | SQuAD-Both-Cut-R1 | **79.92 (+2.62)** | 78.76 (+1.71) | 79.61 (+1.94) | 79.64 (+2.20) |

high signal-to-noise ratio information (rather than incorrectly covering noise), and its effectiveness is significantly enhanced. `TopK`-based methods are more likely to "cover" high-quality information segments truly relevant to the answer, whereas `Div`, as an intrinsic metric, inherently includes effective mechanisms not directly dependent on precise semantic coverage (e.g., structural diversity: selecting examples with different sentence structures or argumentation styles). These mechanisms already play a role in the original noisy data, avoiding overfitting to noise, causing it to outperform noise-sensitive coverage strategies, and its baseline performance is already relatively robust. This fully demonstrates that the value of *diversity* is "beyond coverage".

# D  ADDITIONAL ABLATION STUDIES

## D.1  RESULTS ON MORE MODELS

We evaluated different model sizes from the Gemma and Llama families, including Llama-3.2-1B, Gemma-2-2B, Llama-3.2-3B, Llama-3.1-8B, Gemma-2-9B, Gemma-2-27B, and Llama-3.1-70B. For math tasks, we used the instruct version of the corresponding models. For other tasks, we used the base models. For code tasks, we also tested domain-specific CodeLlama models, including CodeLlama-7B-hf, CodeLlama-13B-hf, and CodeLlama-34B-hf. The results on CodeLlama were consistent with those of other base models.

We report the complete experimental results of the Llama family in Table 11, the Gemma family results in Table 12, and the CodeLlama results in Table 13. The methods that performed well on the corresponding tasks in Table 1 also demonstrated good performance across different model sizes.

Table 10: We supplemented the content omitted in Table 1. The main numerical values represent the mean results over ten random seeds, while the subscript indicates their std. We still highlight the result with the highest mean in bold. In most cases, the fluctuations within each method do not affect our conclusions.

| Model | K | Method | Classification | | Multiple-choice | | Math | | Code | Reading | |
|---|---|---|---|---|---|---|---|---|---|---|---|
| | | | SST-2 | Amazon | ARC-Easy | CsQA | GSM8K | GSM-Plus-Mini | GeoQuery | SQuAD | SCIQ |
| Llama-3.1-8B | 0 | - | 87.50 | 95.40 | 82.43 | 62.80 | 53.45 | 65.12 | — | 42.30 | 36.40 |
| | 4 | Rand | $91.31_{0.59}$ | $\mathbf{96.38_{0.25}}$ | $84.72_{0.35}$ | $71.15_{0.65}$ | $\mathbf{82.24_{0.52}}$ | $66.90_{0.59}$ | $12.57_{1.33}$ | $\mathbf{75.95_{0.55}}$ | $74.00_{0.57}$ |
| | | TopK | $\mathbf{94.13_{0.21}}$ | $96.24_{0.16}$ | $\mathbf{86.10_{0.32}}$ | $72.54_{0.36}$ | $81.99_{0.55}$ | $65.30_{0.46}$ | $63.04_{1.96}$ | $73.51_{0.48}$ | $72.70_{0.40}$ |
| | | Div | $91.50_{0.63}$ | $96.18_{0.25}$ | $85.06_{0.27}$ | $71.17_{0.42}$ | $82.14_{0.45}$ | $\mathbf{66.92_{0.52}}$ | $33.71_{1.35}$ | $75.66_{0.97}$ | $\mathbf{74.47_{0.62}}$ |
| | | TopK-Div | $92.75_{0.33}$ | $96.15_{0.22}$ | $85.83_{0.38}$ | $\mathbf{72.57_{0.35}}$ | $81.74_{0.53}$ | $66.12_{0.85}$ | $\mathbf{71.07_{1.11}}$ | $73.28_{0.79}$ | $73.87_{0.35}$ |
| | 8 | Rand | $92.27_{0.55}$ | $\mathbf{96.63_{0.27}}$ | $84.38_{0.34}$ | $72.23_{0.34}$ | $82.81_{0.61}$ | $\mathbf{66.72_{0.72}}$ | $23.21_{1.41}$ | $77.13_{0.80}$ | $74.65_{0.88}$ |
| | | TopK | $\mathbf{93.64_{0.36}}$ | $96.12_{0.09}$ | $\mathbf{85.91_{0.29}}$ | $\mathbf{73.91_{0.38}}$ | $82.26_{0.65}$ | $65.99_{0.60}$ | $72.04_{0.93}$ | $75.52_{0.43}$ | $74.72_{0.65}$ |
| | | Div | $92.95_{0.35}$ | $96.25_{0.19}$ | $84.97_{0.32}$ | $72.77_{0.61}$ | $\mathbf{82.98_{0.34}}$ | $66.56_{0.60}$ | $38.54_{0.90}$ | $\mathbf{77.71_{0.80}}$ | $\mathbf{75.17_{0.53}}$ |
| | | TopK-Div | $93.33_{0.36}$ | $96.43_{0.09}$ | $85.39_{0.40}$ | $73.76_{0.37}$ | $82.63_{0.57}$ | $66.48_{0.52}$ | $\mathbf{78.36_{1.24}}$ | $76.13_{0.42}$ | $75.07_{0.49}$ |
| Gemma-2-9B | 0 | - | 67.50 | 85.10 | 88.15 | 61.80 | 16.07 | 32.79 | — | 37.90 | 41.10 |
| | 4 | Rand | $93.33_{0.52}$ | $96.15_{0.23}$ | $89.52_{0.25}$ | $74.70_{0.70}$ | $84.29_{0.43}$ | $74.40_{0.47}$ | $13.89_{1.67}$ | $\mathbf{77.19_{0.89}}$ | $75.80_{0.54}$ |
| | | TopK | $\mathbf{94.47_{0.48}}$ | $96.34_{0.20}$ | $\mathbf{90.50_{0.16}}$ | $75.19_{0.25}$ | $84.25_{0.73}$ | $\mathbf{74.50_{0.55}}$ | $61.14_{1.33}$ | $74.82_{0.70}$ | $75.24_{0.34}$ |
| | | Div | $93.45_{0.46}$ | $95.69_{0.23}$ | $90.03_{0.24}$ | $74.85_{0.39}$ | $\mathbf{84.44_{0.91}}$ | $73.34_{0.62}$ | $36.29_{1.05}$ | $77.06_{0.57}$ | $\mathbf{75.96_{0.55}}$ |
| | | TopK-Div | $93.34_{0.34}$ | $\mathbf{96.57_{0.16}}$ | $90.19_{0.19}$ | $\mathbf{75.60_{0.54}}$ | $83.54_{0.56}$ | $74.47_{0.63}$ | $\mathbf{70.43_{1.24}}$ | $75.05_{0.41}$ | $75.21_{0.29}$ |
| | 8 | Rand | $93.30_{0.36}$ | $96.09_{0.23}$ | $89.39_{0.28}$ | $75.98_{0.56}$ | $\mathbf{84.34_{0.54}}$ | $74.48_{0.63}$ | $24.36_{1.19}$ | $\mathbf{79.23_{0.64}}$ | $76.28_{0.50}$ |
| | | TopK | $\mathbf{94.20_{0.28}}$ | $96.55_{0.16}$ | $\mathbf{90.62_{0.16}}$ | $76.14_{0.63}$ | $83.57_{0.53}$ | $\mathbf{75.36_{0.43}}$ | $71.00_{1.20}$ | $77.59_{0.42}$ | $75.55_{0.18}$ |
| | | Div | $93.41_{0.20}$ | $95.94_{0.25}$ | $89.90_{0.19}$ | $\mathbf{76.60_{0.32}}$ | $84.22_{0.52}$ | $74.69_{0.64}$ | $42.07_{1.10}$ | $79.05_{0.93}$ | $\mathbf{76.65_{0.60}}$ |
| | | TopK-Div | $94.04_{0.29}$ | $\mathbf{96.58_{0.04}}$ | $90.48_{0.22}$ | $76.53_{0.21}$ | $83.85_{0.66}$ | $75.16_{0.32}$ | $\mathbf{76.32_{0.85}}$ | $77.64_{0.63}$ | $76.24_{0.48}$ |
| Mistral-7B-v0.3 | 0 | - | 66.50 | 94.00 | 76.41 | 51.80 | 9.48 | 5.17 | — | 30.50 | 34.20 |
| | 4 | Rand | $91.00_{0.78}$ | $94.02_{0.61}$ | $82.77_{0.48}$ | $69.83_{0.81}$ | $48.78_{1.00}$ | $37.20_{0.69}$ | $12.14_{1.47}$ | $\mathbf{76.70_{0.72}}$ | $74.71_{0.54}$ |
| | | TopK | $\mathbf{93.57_{0.25}}$ | $\mathbf{96.17_{0.20}}$ | $\mathbf{85.21_{0.30}}$ | $69.73_{0.43}$ | $49.28_{1.17}$ | $38.20_{0.55}$ | $60.14_{0.82}$ | $75.04_{0.74}$ | $73.73_{0.59}$ |
| | | Div | $91.98_{0.46}$ | $94.15_{0.31}$ | $82.98_{0.25}$ | $70.15_{0.56}$ | $49.49_{0.87}$ | $37.50_{0.76}$ | $34.89_{1.39}$ | $75.96_{1.08}$ | $\mathbf{75.83_{0.57}}$ |
| | | TopK-Div | $92.73_{0.30}$ | $95.90_{0.15}$ | $84.55_{0.20}$ | $69.91_{0.49}$ | $\mathbf{49.99_{1.02}}$ | $\mathbf{38.45_{0.81}}$ | $\mathbf{71.46_{1.35}}$ | $74.43_{0.50}$ | $73.16_{0.28}$ |
| | 8 | Rand | $92.49_{0.34}$ | $95.35_{0.36}$ | $83.69_{0.36}$ | $71.65_{0.60}$ | $47.86_{1.19}$ | $36.32_{0.71}$ | $22.18_{1.96}$ | $77.30_{0.54}$ | $75.54_{0.63}$ |
| | | TopK | $\mathbf{93.61_{0.32}}$ | $\mathbf{96.15_{0.16}}$ | $\mathbf{85.17_{0.25}}$ | $71.88_{0.38}$ | $48.43_{1.02}$ | $37.35_{0.53}$ | $50.51_{0.36}$ | $77.05_{0.41}$ | $75.44_{0.43}$ |
| | | Div | $92.55_{0.29}$ | $95.10_{0.37}$ | $84.27_{0.41}$ | $\mathbf{72.04_{0.61}}$ | $48.33_{1.10}$ | $36.12_{0.34}$ | $39.14_{1.40}$ | $\mathbf{77.67_{1.56}}$ | $\mathbf{76.30_{0.31}}$ |
| | | TopK-Div | $93.47_{0.41}$ | $96.11_{0.16}$ | $84.85_{0.34}$ | $71.81_{0.19}$ | $\mathbf{48.60_{0.71}}$ | $\mathbf{37.81_{0.76}}$ | $\mathbf{77.93_{1.70}}$ | $77.44_{0.37}$ | $75.22_{0.42}$ |

Table 11: Performance of different algorithms for models belong to Llama-family. Setting same as Table 1 while adding results from more models (Llama-3.2-1B, Llama-3.2-3B, Llama-3.1-70B). Our finding that diversity helps for more challenging tasks still holds.

| model | K | Method | Classification | | Multiple-choice | | Math | | Code | Reading | |
|---|---|---|---|---|---|---|---|---|---|---|---|
| | | | SST-2 | Amazon | Arc-easy | CsQA | GSM8K | GSM-Plus-Mini | GeoQuery | SQuAD | SCIQ |
| Llama-3.2-1B | 4 | Rand | $86.88_{0.49}$ | $90.64_{0.43}$ | $71.65_{0.46}$ | $59.54_{0.63}$ | $\mathbf{22.18_{0.81}}$ | $12.62_{0.56}$ | $7.43_{1.27}$ | $56.17_{0.75}$ | $62.75_{0.84}$ |
| | | TopK | $\mathbf{91.87_{0.49}}$ | $\mathbf{93.46_{0.25}}$ | $\mathbf{75.28_{0.57}}$ | $60.20_{0.46}$ | $19.57_{0.71}$ | $9.41_{0.60}$ | $48.25_{0.94}$ | $55.09_{0.38}$ | $\mathbf{63.65_{0.36}}$ |
| | | Div | $88.35_{1.19}$ | $91.14_{0.43}$ | $73.52_{0.54}$ | $60.60_{0.65}$ | $21.29_{1.05}$ | $\mathbf{13.16_{0.73}}$ | $29.46_{1.70}$ | $55.40_{1.18}$ | $63.38_{0.54}$ |
| | | TopK-Div | $91.47_{0.60}$ | $93.22_{0.33}$ | $74.62_{0.44}$ | $\mathbf{60.89_{0.36}}$ | $20.30_{0.80}$ | $9.35_{0.53}$ | $\mathbf{56.57_{1.18}}$ | $\mathbf{56.39_{0.96}}$ | $62.96_{0.31}$ |
| | 8 | Rand | $89.56_{0.81}$ | $92.62_{0.39}$ | $72.72_{0.32}$ | $61.27_{0.62}$ | $21.04_{0.63}$ | $10.01_{0.44}$ | $13.04_{1.71}$ | $\mathbf{58.76_{0.56}}$ | $65.05_{0.45}$ |
| | | TopK | $\mathbf{92.91_{0.30}}$ | $93.95_{0.24}$ | $\mathbf{75.41_{0.40}}$ | $61.77_{0.35}$ | $16.58_{0.53}$ | $7.12_{0.44}$ | $56.29_{1.94}$ | $58.38_{0.66}$ | $65.87_{0.45}$ |
| | | Div | $87.96_{1.14}$ | $92.72_{0.34}$ | $73.53_{0.45}$ | $\mathbf{62.24_{0.55}}$ | $\mathbf{22.24_{0.94}}$ | $11.73_{1.55}$ | $32.75_{1.46}$ | $58.37_{1.16}$ | $65.89_{0.94}$ |
| | | TopK-Div | $92.30_{0.38}$ | $\mathbf{94.06_{0.20}}$ | $74.74_{0.28}$ | $62.20_{0.50}$ | $16.00_{0.81}$ | $6.45_{0.46}$ | $\mathbf{65.11_{1.63}}$ | $58.27_{0.74}$ | $\mathbf{66.24_{0.57}}$ |
| Llama-3.2-3B | 4 | Rand | $90.40_{0.66}$ | $95.87_{0.21}$ | $78.62_{0.44}$ | $68.51_{0.64}$ | $69.64_{0.98}$ | $50.50_{0.59}$ | $9.86_{1.28}$ | $\mathbf{71.59_{0.60}}$ | $72.43_{0.70}$ |
| | | TopK | $92.87_{0.31}$ | $96.25_{0.24}$ | $81.51_{0.34}$ | $68.72_{0.37}$ | $\mathbf{70.05_{0.86}}$ | $50.10_{0.53}$ | $54.04_{1.68}$ | $71.21_{0.48}$ | $70.87_{0.46}$ |
| | | Div | $90.87_{0.59}$ | $95.57_{0.18}$ | $80.41_{0.60}$ | $68.80_{0.57}$ | $68.71_{0.75}$ | $\mathbf{51.53_{0.87}}$ | $31.21_{1.82}$ | $71.13_{1.42}$ | $\mathbf{72.58_{0.61}}$ |
| | | TopK-Div | $\mathbf{93.03_{0.29}}$ | $\mathbf{96.43_{0.15}}$ | $\mathbf{81.71_{0.30}}$ | $\mathbf{68.95_{0.25}}$ | $69.14_{0.86}$ | $50.60_{0.38}$ | $\mathbf{59.75_{2.03}}$ | $70.73_{0.49}$ | $71.28_{0.40}$ |
| | 8 | Rand | $91.79_{0.36}$ | $96.09_{0.14}$ | $78.91_{0.40}$ | $69.89_{0.68}$ | $68.79_{0.94}$ | $\mathbf{51.12_{0.81}}$ | $19.29_{1.60}$ | $73.14_{0.63}$ | $72.57_{0.85}$ |
| | | TopK | $\mathbf{93.71_{0.40}}$ | $96.18_{0.20}$ | $81.03_{0.36}$ | $70.53_{0.33}$ | $\mathbf{69.40_{0.89}}$ | $50.00_{0.84}$ | $61.89_{1.49}$ | $72.61_{0.30}$ | $71.83_{0.46}$ |
| | | Div | $91.99_{0.47}$ | $95.83_{0.27}$ | $80.68_{0.34}$ | $70.26_{0.40}$ | $66.54_{1.14}$ | $50.87_{0.61}$ | $36.71_{1.39}$ | $72.76_{1.53}$ | $\mathbf{74.07_{0.49}}$ |
| | | TopK-Div | $93.55_{0.35}$ | $\mathbf{96.37_{0.17}}$ | $\mathbf{81.57_{0.30}}$ | $70.18_{0.33}$ | $69.38_{0.98}$ | $49.96_{0.73}$ | $\mathbf{72.11_{1.77}}$ | $\mathbf{73.80_{0.56}}$ | $72.08_{0.53}$ |
| Llama-3.1-8B | 4 | Rand | $91.31_{0.59}$ | $\mathbf{96.38_{0.25}}$ | $84.72_{0.35}$ | $71.15_{0.65}$ | $\mathbf{82.24_{0.52}}$ | $66.90_{0.59}$ | $12.57_{1.33}$ | $\mathbf{75.95_{0.55}}$ | $74.00_{0.57}$ |
| | | TopK | $\mathbf{94.13_{0.21}}$ | $96.24_{0.16}$ | $\mathbf{86.10_{0.32}}$ | $72.54_{0.36}$ | $81.99_{0.55}$ | $65.30_{0.46}$ | $63.04_{1.96}$ | $73.51_{0.48}$ | $72.70_{0.40}$ |
| | | Div | $91.50_{0.63}$ | $96.18_{0.25}$ | $85.06_{0.27}$ | $71.17_{0.42}$ | $82.14_{0.45}$ | $\mathbf{66.92_{0.52}}$ | $33.71_{1.35}$ | $75.66_{0.97}$ | $\mathbf{74.47_{0.62}}$ |
| | | TopK-Div | $92.75_{0.33}$ | $96.15_{0.22}$ | $85.83_{0.38}$ | $\mathbf{72.57_{0.35}}$ | $81.74_{0.53}$ | $66.12_{0.85}$ | $\mathbf{71.07_{1.11}}$ | $73.28_{0.79}$ | $73.87_{0.35}$ |
| | 8 | Rand | $92.27_{0.55}$ | $\mathbf{96.63_{0.27}}$ | $84.38_{0.34}$ | $72.23_{0.34}$ | $82.81_{0.61}$ | $\mathbf{66.72_{0.72}}$ | $23.21_{1.41}$ | $77.13_{0.80}$ | $74.65_{0.88}$ |
| | | TopK | $\mathbf{93.64_{0.32}}$ | $96.12_{0.09}$ | $\mathbf{85.91_{0.29}}$ | $\mathbf{73.91_{0.38}}$ | $82.26_{0.65}$ | $65.99_{0.60}$ | $72.04_{0.93}$ | $75.52_{0.43}$ | $74.72_{0.65}$ |
| | | Div | $92.95_{0.35}$ | $96.25_{0.19}$ | $84.97_{0.32}$ | $72.77_{0.61}$ | $\mathbf{82.98_{0.34}}$ | $66.56_{0.60}$ | $38.54_{0.90}$ | $\mathbf{77.71_{0.80}}$ | $\mathbf{75.17_{0.53}}$ |
| | | TopK-Div | $93.33_{0.36}$ | $96.43_{0.09}$ | $85.39_{0.40}$ | $73.76_{0.37}$ | $82.63_{0.57}$ | $66.48_{0.52}$ | $\mathbf{78.36_{1.24}}$ | $76.13_{0.42}$ | $75.07_{0.49}$ |
| Llama-3.1-70B | 4 | Rand | $94.16_{0.33}$ | $96.77_{0.38}$ | $89.76_{0.16}$ | $75.48_{0.62}$ | $88.64_{0.48}$ | $77.14_{0.39}$ | $17.50_{1.88}$ | $\mathbf{81.47_{0.75}}$ | $75.51_{0.87}$ |
| | | TopK | $\mathbf{94.81_{0.34}}$ | $96.86_{0.11}$ | $\mathbf{90.57_{0.24}}$ | $\mathbf{76.22_{0.28}}$ | $88.87_{0.52}$ | $76.19_{0.57}$ | $66.46_{1.23}$ | $79.15_{0.22}$ | $75.67_{0.38}$ |
| | | Div | $94.34_{0.28}$ | $96.36_{0.23}$ | $90.14_{0.30}$ | $75.53_{0.33}$ | $\mathbf{89.27_{0.53}}$ | $\mathbf{77.21_{0.47}}$ | $39.00_{1.87}$ | $81.27_{1.25}$ | $\mathbf{77.75_{0.44}}$ |
| | | TopK-Div | $94.20_{0.18}$ | $\mathbf{96.88_{0.12}}$ | $90.46_{0.32}$ | $76.21_{0.49}$ | $88.67_{0.42}$ | $76.94_{0.45}$ | $\mathbf{77.32_{0.95}}$ | $79.26_{0.37}$ | $75.59_{0.33}$ |
| | 8 | Rand | $94.66_{0.36}$ | $96.95_{0.28}$ | $89.84_{0.29}$ | $77.14_{0.40}$ | $89.47_{0.54}$ | $76.93_{0.77}$ | $26.89_{1.90}$ | $82.62_{0.47}$ | $76.56_{0.78}$ |
| | | TopK | $94.18_{0.32}$ | $96.95_{0.18}$ | $90.24_{0.25}$ | $\mathbf{77.65_{0.30}}$ | $89.33_{0.29}$ | $76.29_{0.46}$ | $75.68_{1.08}$ | $81.38_{0.51}$ | $76.70_{0.63}$ |
| | | Div | $\mathbf{94.95_{0.30}}$ | $96.47_{0.25}$ | $89.99_{0.17}$ | $77.33_{0.59}$ | $\mathbf{89.65_{0.27}}$ | $\mathbf{77.11_{0.29}}$ | $44.25_{1.92}$ | $\mathbf{83.18_{1.43}}$ | $\mathbf{78.37_{0.60}}$ |
| | | TopK-Div | $94.75_{0.19}$ | $\mathbf{97.27_{0.11}}$ | $\mathbf{90.71_{0.25}}$ | $77.24_{0.31}$ | $89.17_{0.42}$ | $76.74_{0.92}$ | $\mathbf{81.39_{1.16}}$ | $81.47_{0.23}$ | $76.77_{0.41}$ |

Due to resource constraints, our experiments primarily focused on mainstream open-source models. We tested the Math task on the commercial-grade models gpt-4o-mini and deepseek-v3. As shown in Table 14, the Div method consistently outperformed TopK on both gsm8k and prm800k.

Table 12: Performance of different algorithms for models belong to Gemma-family. Setting same as Table 1 while adding results from more models (Gemma-2-2b and Gemma-2-27b). Our finding that diversity helps for more challenging tasks still holds.

| model | $K$ | Method | Classification | | Multiple-choice | | Math | | Code | Reading | |
|---|---|---|---|---|---|---|---|---|---|---|---|
| | | | SST-2 | Amazon | Arc-easy | CsQA | GSM8K | GSM-Plus-Mini | GeoQuery | SQuAD | SCIQ |
| Gemma-2-2B | 4 | Rand | $85.00_{0.94}$ | $92.34_{0.75}$ | $82.84_{0.45}$ | $68.89_{0.61}$ | $40.45_{1.11}$ | $33.43_{0.69}$ | $9.14_{1.38}$ | $\mathbf{69.03_{0.34}}$ | $71.27_{0.63}$ |
| | | TopK | $90.67_{0.55}$ | $\mathbf{95.14_{0.18}}$ | $\mathbf{84.57_{0.35}}$ | $69.93_{0.47}$ | $41.33_{0.85}$ | $\mathbf{34.39_{0.71}}$ | $53.39_{1.88}$ | $68.19_{0.76}$ | $71.09_{0.49}$ |
| | | Div | $89.63_{0.54}$ | $92.55_{0.44}$ | $84.21_{0.37}$ | $\mathbf{71.03_{0.62}}$ | $40.53_{2.15}$ | $32.75_{1.64}$ | $31.29_{1.47}$ | $67.73_{1.18}$ | $\mathbf{72.34_{0.45}}$ |
| | | TopK-Div | $\mathbf{91.66_{0.57}}$ | $95.01_{0.22}$ | $84.51_{0.32}$ | $70.72_{0.52}$ | $\mathbf{42.74_{0.57}}$ | $34.39_{0.64}$ | $\mathbf{61.04_{1.55}}$ | $67.85_{0.60}$ | $71.64_{0.30}$ |
| | 8 | Rand | $89.96_{0.51}$ | $94.10_{0.47}$ | $82.62_{0.33}$ | $70.26_{0.34}$ | $36.44_{0.82}$ | $34.55_{0.46}$ | $16.68_{2.01}$ | $69.99_{0.68}$ | $72.12_{0.29}$ |
| | | TopK | $92.22_{0.46}$ | $\mathbf{95.58_{0.23}}$ | $84.30_{0.19}$ | $71.06_{0.44}$ | $\mathbf{43.05_{0.69}}$ | $36.45_{0.42}$ | $61.00_{1.44}$ | $69.15_{0.44}$ | $\mathbf{72.35_{0.44}}$ |
| | | Div | $91.81_{0.48}$ | $94.57_{0.37}$ | $84.37_{0.38}$ | $\mathbf{72.22_{0.62}}$ | $38.87_{1.28}$ | $34.07_{1.12}$ | $35.29_{1.25}$ | $69.31_{0.99}$ | $72.28_{0.43}$ |
| | | TopK-Div | $\mathbf{92.40_{0.28}}$ | $95.53_{0.20}$ | $\mathbf{84.39_{0.24}}$ | $71.99_{0.36}$ | $42.93_{0.66}$ | $\mathbf{36.47_{0.47}}$ | $\mathbf{68.93_{1.35}}$ | $\mathbf{70.09_{0.54}}$ | $72.30_{0.37}$ |
| Gemma-2-9B | 4 | Rand | $93.33_{0.52}$ | $96.15_{0.23}$ | $89.52_{0.25}$ | $74.70_{0.70}$ | $84.29_{0.43}$ | $74.40_{0.47}$ | $13.89_{1.67}$ | $\mathbf{77.19_{0.89}}$ | $75.80_{0.54}$ |
| | | TopK | $\mathbf{94.47_{0.48}}$ | $96.34_{0.20}$ | $\mathbf{90.50_{0.16}}$ | $75.19_{0.25}$ | $84.25_{0.73}$ | $\mathbf{74.50_{0.55}}$ | $61.14_{1.33}$ | $74.82_{0.70}$ | $75.24_{0.34}$ |
| | | Div | $93.45_{0.46}$ | $95.69_{0.23}$ | $90.03_{0.24}$ | $74.85_{0.39}$ | $\mathbf{84.44_{0.91}}$ | $73.34_{0.62}$ | $36.29_{1.05}$ | $77.06_{0.57}$ | $\mathbf{75.96_{0.55}}$ |
| | | TopK-Div | $93.34_{0.34}$ | $\mathbf{96.57_{0.16}}$ | $90.19_{0.19}$ | $\mathbf{75.60_{0.54}}$ | $83.54_{0.56}$ | $74.47_{0.63}$ | $\mathbf{70.43_{1.24}}$ | $75.05_{0.41}$ | $75.21_{0.29}$ |
| | 8 | Rand | $93.30_{0.36}$ | $96.09_{0.23}$ | $89.39_{0.28}$ | $75.98_{0.56}$ | $\mathbf{84.34_{0.54}}$ | $74.48_{0.63}$ | $24.36_{1.19}$ | $\mathbf{79.23_{0.64}}$ | $76.28_{0.50}$ |
| | | TopK | $\mathbf{94.20_{0.28}}$ | $96.55_{0.16}$ | $\mathbf{90.62_{0.16}}$ | $76.14_{0.63}$ | $83.57_{0.53}$ | $\mathbf{75.36_{0.43}}$ | $71.00_{1.20}$ | $77.59_{0.42}$ | $75.55_{0.18}$ |
| | | Div | $93.41_{0.20}$ | $95.94_{0.25}$ | $89.90_{0.19}$ | $\mathbf{76.60_{0.32}}$ | $84.22_{0.52}$ | $74.69_{0.64}$ | $42.07_{1.10}$ | $79.05_{0.93}$ | $\mathbf{76.65_{0.60}}$ |
| | | TopK-Div | $94.04_{0.29}$ | $\mathbf{96.58_{0.04}}$ | $90.48_{0.22}$ | $76.53_{0.21}$ | $83.85_{0.66}$ | $75.16_{0.32}$ | $\mathbf{76.32_{0.85}}$ | $77.64_{0.63}$ | $76.24_{0.48}$ |
| Gemma-2-27B | 4 | Rand | $94.16_{0.40}$ | $96.06_{0.26}$ | $\mathbf{89.99_{0.26}}$ | $76.15_{0.41}$ | $90.16_{0.33}$ | $\mathbf{70.76_{0.71}}$ | $18.68_{1.83}$ | $\mathbf{80.54_{0.59}}$ | $75.61_{0.78}$ |
| | | TopK | $\mathbf{95.00_{0.33}}$ | $96.47_{0.11}$ | $89.64_{0.15}$ | $76.47_{0.46}$ | $89.73_{0.38}$ | $69.27_{0.49}$ | $67.75_{1.45}$ | $78.43_{0.49}$ | $76.35_{0.46}$ |
| | | Div | $94.15_{0.43}$ | $95.57_{0.36}$ | $89.78_{0.40}$ | $\mathbf{76.97_{0.47}}$ | $\mathbf{90.68_{0.23}}$ | $69.85_{1.60}$ | $41.75_{2.16}$ | $79.91_{1.14}$ | $\mathbf{76.73_{0.61}}$ |
| | | TopK-Div | $94.43_{0.18}$ | $\mathbf{96.59_{0.12}}$ | $89.76_{0.16}$ | $76.15_{0.41}$ | $89.53_{0.23}$ | $69.62_{0.56}$ | $\mathbf{79.11_{0.97}}$ | $78.39_{0.46}$ | $75.91_{0.53}$ |
| | 8 | Rand | $94.59_{0.45}$ | $96.42_{0.46}$ | $89.62_{0.21}$ | $77.17_{0.69}$ | $90.23_{0.25}$ | $69.04_{0.53}$ | $30.36_{2.04}$ | $\mathbf{81.81_{0.37}}$ | $77.09_{0.53}$ |
| | | TopK | $94.30_{0.32}$ | $\mathbf{96.60_{0.18}}$ | $90.14_{0.24}$ | $77.20_{0.42}$ | $89.95_{0.27}$ | $66.54_{0.37}$ | $77.54_{1.00}$ | $80.72_{0.38}$ | $76.42_{0.52}$ |
| | | Div | $\mathbf{94.61_{0.29}}$ | $95.95_{0.27}$ | $90.21_{0.23}$ | $\mathbf{78.47_{0.41}}$ | $\mathbf{90.45_{0.22}}$ | $\mathbf{70.02_{0.91}}$ | $46.96_{2.23}$ | $81.38_{0.89}$ | $\mathbf{77.84_{0.51}}$ |
| | | TopK-Div | $94.56_{0.29}$ | $96.43_{0.13}$ | $\mathbf{90.34_{0.22}}$ | $77.29_{0.31}$ | $89.70_{0.25}$ | $68.48_{0.38}$ | $\mathbf{82.39_{1.35}}$ | $80.90_{0.53}$ | $77.10_{0.54}$ |

Table 13: **CodeLlama-family results on GeoQuery dataset with different split.** We observe that on GeoQuery dataset, TopK-Div consistently works better than TopK, and there is also a large gap between TopK and more diversity-aware methods like Div and Rand, which aligns with the results in Table 11, Table 12, and Table 1 for Mistral-v0.3. The gap between different methods is wide and std is small, so we omit the std.

| model | $K$ | Method | GeoQuery | | | |
|---|---|---|---|---|---|---|
| | | | Standard | Tmcd | Template | Length |
| CodeLlama-7B-hf | 4 | Rand | 12.21 | 10.43 | 9.75 | 3.61 |
| | | TopK | 57.86 | 35.68 | 36.90 | 25.91 |
| | | Div | 33.11 | 21.95 | 27.22 | 13.16 |
| | | TopK-Div | **67.86** | **40.00** | **50.34** | **33.64** |
| | 8 | Rand | 21.11 | 17.25 | 17.93 | 8.05 |
| | | TopK | 58.21 | 42.95 | 48.06 | 32.95 |
| | | Div | 38.29 | 24.75 | 31.41 | 16.48 |
| | | TopK-Div | **66.79** | **46.36** | **55.13** | **39.09** |
| CodeLlama-13B-hf | 4 | Rand | 13.82 | 11.66 | 11.73 | 4.11 |
| | | TopK | 63.57 | 37.73 | 38.04 | 29.77 |
| | | Div | 37.43 | 23.23 | 26.51 | 18.07 |
| | | TopK-Div | **72.14** | **44.32** | **53.99** | **40.68** |
| | 8 | Rand | 24.89 | 18.64 | 21.16 | 9.52 |
| | | TopK | 69.64 | 44.09 | 56.04 | 41.14 |
| | | Div | 42.71 | 26.00 | 30.59 | 20.68 |
| | | TopK-Div | **79.29** | **47.73** | **64.24** | **44.32** |
| CodeLlama-34B-hf | 4 | Rand | 15.75 | 13.02 | 14.42 | 5.98 |
| | | TopK | 63.57 | 42.05 | 43.51 | 30.23 |
| | | Div | 39.86 | 24.75 | 32.92 | 19.18 |
| | | TopK-Div | **72.50** | **48.18** | **56.72** | **44.55** |
| | 8 | Rand | 25.46 | 20.50 | 24.76 | 11.73 |
| | | TopK | 73.93 | 48.41 | 56.04 | 44.09 |
| | | Div | 44.18 | 27.50 | 39.29 | 24.32 |
| | | TopK-Div | **80.71** | **50.00** | **64.92** | **48.86** |

Table 14: Results of GPT-4o-mini and Deepseek-v3 in Math Task.

| Model | K | Dataset | Method | | | |
|---|---|---|---|---|---|---|
| | | | Rand | TopK | Div | TopK-Div |
| GPT-4o-mini | 4 | GSM8K | **93.03** | 91.06 | 92.80 | 92.27 |
| | | PRM800K | 68.40 | 66.60 | **71.20** | 69.20 |
| Deepseek-v3 | 4 | GSM8K | **96.13** | 95.75 | 95.91 | 95.45 |
| | | PRM800K | 85.00 | **87.00** | 85.00 | 86.80 |

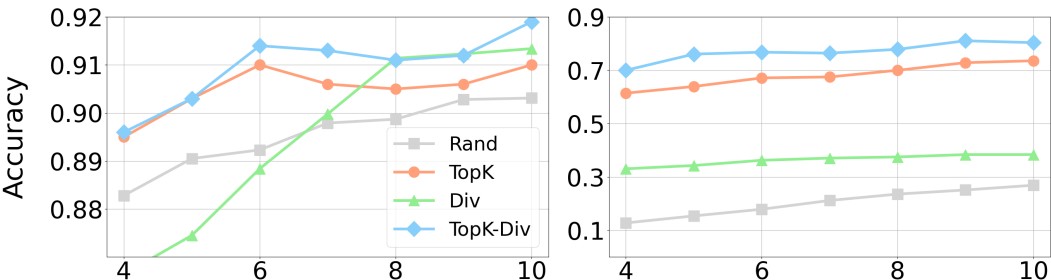

Figure 5: The performance of different demonstration selection methods with different number of shots $K$. **Left:** sentiment classification task with demonstrations come from Amazon and queries come from SST-2. **Right:** text to SQL task with demonstrations and query come from the training and test set of GeoQuery Standard Split.

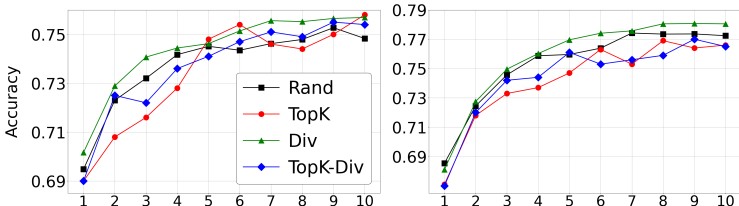

Figure 6: **(The accuracy of different methods with different number of shots $K$ on reading comprehension tasks.)** We choose report the results on Llama-3.1-8B, with Sentence-BERT embeddings (all-roberta-large-v1). **Left:** results where demonstration and query come from SCIQ. **Right:** results where demonstration and query come from SQuAD.

### D.2 Changing the number of shots

In this section, we investigate how the performance advantage of diversity-aware methods over TopK evolves with increasing shot count. Our results in Figure 5 show that the improvement from diversity-aware selection (Div) remains substantial even with higher number of shots.

We believe that as the shot number increases, there is an increase in redundant information among the examples selected by the `TopK` method. In contrast, the `TopK-Div` method minimizes the occurrence of redundant information as much as possible, thereby enabling the model to more clearly identify the task theme.

Figure 5 presents the relative improvement on the GeoQuery standard split and SCIQ — two tasks where diversity-aware methods showed clear benefits (Section 3.1) — across different sizes of the Llama-3.1/3.2 and Gemma-2 model families. The results indicate that, in general, the relative improvement from diversity-aware selection does not diminish significantly as model size increases. This underscores the continued importance of understanding diversity's role in demonstration selection.

We present the experimental results on reading comprehension task (SCIQ, SQuAD), where diversity-aware methods perform well, with different numbers of shots $K$ ranging from 1 to 10. We test different methods on the Llama-3.1-8B model. Figure 6 summarizes. To our surprise, when $k = 1$, `TopK` performs significantly worse than `Rand` on both datasets, indicating that the accuracy of these

Table 15: Results of K-Means Method in Math Task. We add the K-Means baseline based on Table 1 in our paper. The implementation of K-Means consists of two steps: First, partition the input into $k$ clusters using the $k$-Means method. Second, select $k$ points as demonstrations by choosing the point closest to the cluster center within each cluster.

| model | $K$ | Dataset | Method | | | | |
| | | | Rand | Topk | Div | Topk-Div | K-means |
|---|---|---|---|---|---|---|---|
| Llama-3.1-8B | 4 | GSM8K | 82.24 | 81.99 | 82.14 | 81.74 | **83.89** |
| | | GSM-Plus-M | 66.90 | 65.30 | 66.92 | 66.12 | **68.10** |
| | 8 | GSM8K | 82.81 | 82.26 | **82.98** | 82.63 | 82.36 |
| | | GSM-Plus-M | 66.72 | 65.99 | 66.56 | 66.48 | 66.52 |
| Gemma-2-9B | 4 | GSM8K | 84.29 | 84.25 | 84.44 | 83.54 | **85.24** |
| | | GSM-Plus-M | 74.40 | 74.50 | 73.34 | 74.47 | 74.52 |
| | 8 | GSM8K | 84.34 | 83.57 | 84.22 | 83.85 | **84.97** |
| | | GSM-Plus-M | 74.48 | 75.36 | 74.69 | 75.16 | **76.29** |
| Mistral-7B-v0.3 | 4 | GSM8K | 48.78 | 49.28 | 49.49 | **49.99** | 43.90 |
| | | GSM-Plus-M | 37.20 | 38.20 | 37.50 | **38.45** | 35.50 |
| | 8 | GSM8K | 47.86 | 48.43 | 48.33 | 48.60 | **49.13** |
| | | GSM-Plus-M | 36.32 | 37.35 | 36.12 | **37.81** | 36.41 |

datasets is not solely related to the coverage of example sets. Under most settings of $k$, `Div` shows significant advantages over `TopK`. Moreover, the correlation between example sets selected by `Div` and test samples is relatively low. This sufficiently demonstrates that even when example samples do not have coverage of test samples, they can still be high-quality examples, which is also consistent with the good performance of `Rand`.

### D.3 MORE DIVERSITY-AWARE METHOD

In the main text, `TopK-Div` and `Div` are both diversity-aware methods that combine the `TopK` method. We want to understand what happens when using a purely diversity-based method. Therefore, we implemented the `K-Means` method: dividing the training set into k clusters by k-means algorithm and then choose the nearest sample to the Centroid from each cluster (`K-Means`), `K-Means` can be viewed as a purely diversity-based met.

The results in Table 15 show that the `K-Means` method still has advantages compared to the `TopK` method. In fact, we believe `Rand` can also be considered a purely diversity-based method. This implies the advantage of diversity methods does not depend on the specific implementation.

### D.4 ABALATIONS ON THE SIZE OF TRAINING SET

To investigate whether the way diversity works is related to the size of the training set—for example, whether the example selection strategy needs to change when the available training set is limited, We conducted experiments on the SQuAD and SCIQ datasets by randomly sampling 50 examples from each training set to create SCIQ-50 and SQuAD-50, while keeping the original testing set unchanged.

When the available training set size is reduced, TopK still underperforms compared to Div, maintaining an average performance gap of 1% in 4-shot and 8-shot settings.

### D.5 ABALATIONS ON "BETTER" EMBEDDINGS

**"better" embedding in a cheating way.** All methods we test, except randomly chosen (`Rand`), depend on an embedding model. It is always possible that the embedding model is not good enough. Indeed, using Sentence-BERT on questions/input (optimized for semantic similarity) might not be optimal for math tasks and text-to-SQL generation, and the ideal embedding might be highly dependent on the structure or reasoning steps of the answer. In this section, we test if diversity still helps when given a better embedding, computed in a "cheating" way: For math problems, we append the gold answer after the question and compute the embedding using Sentence-BERT; For text-to-SQL generation, we compute the occurrence of keywords in the answer (Levy et al., 2023).

Table 16: Embedding on answer using Gemma-2-9B with 4 shots. Comparing to Table 1, the relative ranking between the tested methods doesn't change.

|  | Rand | TopK | Div | TopK-Div |
|---|---|---|---|---|
| GSM8K | 82.21 | 84.53 | 84.14 | **84.69** |
| PRM800K | 38.04 | 45.60 | 37.56 | **46.40** |
| GeoQuery(Standard) | 13.71 | 79.64 | 54.32 | **84.29** |

Table 17: **Results of different embeddings on Llama-3.1-8B.** We test different methods using different similarity scores computation ("all-roberta-large-v1", "BM25", "BertScore"). We test on Llama-3.1-8B model on math (using instruct model) and reading comprehension (using base model). The numbers for embedding "all-roberta-large-v1" are copied from Table 1. The numbers corresponding to Rand for BM25 and BertScore are also copied. We find that: (1) using another embedding might affect the TopK performance, as we can observe an increase of performance for TopK while changing to BM25 or BertScore. (2) Diversity still helps, since if we look at the best performance with the best embedding, in most of the cases the best performance is still achieved by diversity-aware methods.

| Embedding | $K$ | Method | Math | | Reading | |
|---|---|---|---|---|---|---|
|  |  |  | GSM8K | PRM800K | SQuAD | SCIQ |
| all-roberta-large-v1 | 4 | Rand | 82.40 | 43.50 | 75.87 | 74.17 |
|  |  | TopK | **82.64** | 41.40 | 73.70 | 72.80 |
|  |  | Div | 82.43 | **44.86** | **76.02** | **74.44** |
|  |  | TopK-Div | 81.43 | 44.80 | 74.40 | 73.60 |
|  | 8 | Rand | 82.77 | 43.32 | 77.35 | 74.79 |
|  |  | TopK | 82.11 | 43.00 | 76.90 | 74.40 |
|  |  | Div | **83.13** | **44.28** | **78.05** | **75.52** |
|  |  | TopK-Div | 81.73 | 40.00 | 75.90 | 74.90 |
| BM25 | 4 | Rand | 82.40 | 43.50 | 75.87 | 74.17 |
|  |  | TopK | 81.88 | 42.00 | 73.80 | **74.40** |
|  |  | Div | **82.47** | 44.12 | **76.65** | 72.74 |
|  |  | TopK-Div | 81.20 | **45.00** | 75.50 | 74.30 |
|  | 8 | Rand | 82.77 | 43.32 | 77.35 | 74.79 |
|  |  | TopK | 82.94 | 44.80 | 76.60 | **75.20** |
|  |  | Div | **83.44** | 43.92 | **78.97** | 74.12 |
|  |  | TopK-Div | 83.02 | **45.60** | 77.50 | 74.50 |
| BertScore | 4 | Rand | 82.40 | 43.50 | **75.87** | 74.17 |
|  |  | TopK | 81.58 | **45.60** | 75.00 | **74.30** |
|  |  | Div | **82.81** | 44.06 | 74.16 | 73.06 |
|  |  | TopK-Div | 81.05 | 44.40 | 74.90 | 73.20 |
|  | 8 | Rand | 82.77 | 43.32 | **77.35** | 74.79 |
|  |  | TopK | 82.34 | 42.60 | 76.40 | **75.50** |
|  |  | Div | **83.09** | 43.68 | 76.00 | 74.95 |
|  |  | TopK-Div | 81.58 | **44.00** | 75.70 | 74.70 |

Table 16 summarizes the result using the "cheating" embeddings on Gemma-2-9B, and in general, diversity still helps for these tasks.

**Computing local structure for GeoQuery.** For the code-standard task, we tokenized the sample answers at the word level and obtained 52 distinct tokens, with each dimension representing a token. For a given sample, in its 52-dimensional vector, if the corresponding token appears in its answer, the value at that position is 1, otherwise 0. We use this embedding as the code embedding on answers.

**BM25 and BertScore for math and reading comprehension.** We conduct ablation studies on the model to compute the similarity score, changing from cosine similarity from embeddings computed by "all-roberta-large-v1" to BM25 and BertScore, and test different methods on math and reading comprehension tasks. Table 17 summarizes our results. We find that (1) using another embedding

might affect the `TopK` performance, as we can observe an increase in performance for `TopK` while changing to BM25 or BertScore. (2) Diversity still helps since if we look at the best performance with the best embedding, in most cases, the best performance is still achieved by diversity-aware methods.

### D.6 DECODING METHOD

Table 18: Decode performance using Llama-3.1-8B on reading comprehension tasks. The number of shot is fixed as 4.

| Decode | Test. | Rand | TopK | Div | TopK-Div |
|---|---|---|---|---|---|
| Greedy | Squad | 75.87 | 73.70 | **76.02** | 74.40 |
| | Sciq | 74.17 | 72.80 | **74.44** | 73.60 |
| Sampling | Squad | 70.93 | 72.40 | 70.95 | **72.80** |
| | Sciq | 66.86 | 66.70 | 67.08 | **67.70** |

In this part we show some preliminary results on changing the decoding strategy for reading comprehension tasks (SQuAD and CommonsenseQA), since for code and math, greedy decoding is known to perform well. By changing greedy decoding to sampling decoding (topP $= 0.95$, Temperature $= 0.7$), we find that the performance of all tasks drops a lot (Table 18), which justifies our decoding strategy selection.

## E THEORETICAL JUSTIFICATION AND SIMULATIONS

In this section, we give a theoretical justification for combining diversity in demonstration selection for ICL, even if the "embedding" is accurate (Theorems E.2 and E.3). Then we validate the superiority of `TopK-Div` compared to `TopK` in more general settings. In Appendix E.3 and Appendix E.4, we also employ the theoretical framework to conduct detailed simulation experiments.

We consider the linear regression model, where there is a task vector $\theta_{\mathcal{T}} \in \mathbb{R}^d$. The data for this task has embedded input $e \in \mathbb{R}^d$ and output $y = \langle \theta_{\mathcal{T}}, e \rangle$. We also have a demonstration set $D = \{(e_i, y_i)\}_{i=1}^n$ with size $n$, where $y_i = \langle \theta_{\mathcal{T}}, e_i \rangle$ and $e_i$ is drawn from the demonstration distribution $\mathcal{D}_{\mathcal{E}}$. Now given a query $e_q$ drawn from the query distribution $\mathcal{Q}_{\mathcal{E}}$, the goal of demonstration selection is to select a subset $S = \{(e_{j_i}, y_{j_i})\}_{i=1}^K$, such that given the demonstrations $S = \{(e_{j_i}, y_{j_i})\}_{i=1}^K$, the LLM predicts the output close to the gold label $y_q = \langle \theta_{\mathcal{T}}, e_q \rangle$, i.e, $y_q \approx \text{LLM}(S, e_q)$. We make the following assumption on the mechanism of LLM for learning linear regression in-context, given the demonstration $S$ and the query $e_q$.

**Assumption E.1 (ICL for linear regression).** Suppose that the task $\mathcal{T}$ is to predict the value of a linear function $y = \langle \theta_{\mathcal{T}}, e \rangle$ and $K$ demonstrations $S = \{(e_{j_i}, y_{j_i})\}_{i=1}^K$ are selected. Denote $E = [e_{j_1}, \ldots, e_{j_K}]^\top \in \mathbb{R}^{K \times d}$ as the data matrix. Then given a query $e_q$, we assume that the prediction given by the LLM is $y_{\text{pred}} = \langle e_q, E^\dagger E \theta_{\mathcal{T}} \rangle$. Namely, the LLM learns the min-norm solution for the overparameterized linear regression.

By this assumption, the prediction loss of $e_q$ is

$$\text{Loss}(e_q) := (y_{\text{pred}} - \langle \theta_{\mathcal{T}}, e_q \rangle)^2 = \langle \theta_{\mathcal{T}} - E^\dagger E \theta_{\mathcal{T}}, e_q \rangle^2.$$

ICL for linear regression has been extensively studied, empirically and theoretically (Appendix A). Theorem E.1 is also empirically justified, where (Akyürek et al., 2023) observed that after pretraining an autoregressive transformer model on noiseless linear regression tasks, the transformer will learn the min-norm solution for the linear regression in-context if the size of demonstrations $K < d$.

We further assume that the embedding for each data $e \in \{0, 1\}^d$. This is inspired by the theoretical framework that each problem from a specific task contains certain skills (or local structures), and an LLM is able to solve that problem perfectly if the LLM knows all the skills (local structures) and is able to compose the skills (local structures) together (Arora and Goyal, 2023; Yu et al., 2024; Zhao et al., 2025). For example, for a specific math problem related to algebra, the skills required to solve this problem are polynomial multiplication and solving equations, while for another math problem related to geometry, the skills required might be changed to coordinate systems and solving equations.

It is also worth noting that the skill(local structure)-based embedding design also gains empirical success. For example, (Levy et al., 2023; Didolkar et al., 2024) improves semantic parsing and (An et al., 2023b) improves math ability by selecting demonstrations that require similar skills or local structures to the query.

**Example I: Diversity benefits from coverage.** We characterize the demonstration distribution $\mathcal{D}_\mathcal{E}$ and the query distribution $\mathcal{Q}_\mathcal{E}$ below. Let $l \geq 200$ be an even number and let $d = 4l$, where the choice of 200 is to simplify the analysis. Let $\mathcal{D}_\mathcal{E}$ be: Uniformly draw a subset $T_1 \subseteq [2l]$ of size $l/2$ and a subset $T_2 \subseteq \{2l + 1, \ldots, 4l\}$ of size $l/2$, and output $e = e_{T_1 \cup T_2}$, i.e., the $i$-th entry of $e$ is 1 iff $i \in T_1 \cup T_2$. Assume the size $n$ of $D$ is sufficiently large that $D$ covers the entire ground set of $\mathcal{D}_\mathcal{E}$. Let $\mathcal{Q}_\mathcal{E}$ be: Uniformly draw a subset $T \subset [2l]$ of size $l$. We have the following theorem, whose proof can be found in Appendix E.1.

**Theorem E.2** (**Justification example I**). *Suppose each entry of $\theta_\mathcal{T}$ is i.i.d. drawn from the uniform distribution on $[0, 1]$. Let $K = 2$ and $\mathcal{D}_\mathcal{E}, \mathcal{Q}_\mathcal{E}$ be as defined above. For a query $e_q$ drawn from $\mathcal{Q}_\mathcal{E}$, let $L, L'$ denote the expected prediction loss of $e_q$ using* TopK *and* TopK-Div, *respectively, where the randomness comes from $\theta_\mathcal{T}$, $e_q$, and the selection of demonstration examples. Then $L > L'$ for any hyperparameter $\alpha \in (0, 1)$ for* TopK-Div.

Intuitively, the selected two demonstration examples of TopK-Div must cover all non-zero entries of $e_q$, while this property is unlikely to hold for TopK. This demonstrates that adding diversity may increase the coverage of demonstration examples to queries and lead to a lower prediction loss, aligning with the findings in Levy et al. (2023); Gupta et al. (2023); Ye et al. (2023).

**Example II: Diversity is beyond coverage.** We again characterize $\mathcal{D}_\mathcal{E}$ and $\mathcal{Q}_\mathcal{E}$ below. Let $l \geq 3$ be an integer and let $d = 4l$. Let $\mathcal{D}_\mathcal{E}$ be: Uniformly draw a subset $T_1 \subseteq [2l]$ of size $l - 1$ and a subset $T_2 \subseteq \{2l + 1, \ldots, 4l\}$ of size 1, and output $e = e_{T_1 \cup T_2}$. Assume the size $n$ of $D$ is sufficiently large that $D$ covers the entire ground set of $\mathcal{D}_\mathcal{E}$. Let $\mathcal{Q}_\mathcal{E}$ be: Uniformly draw a subset $T \subset [2l]$ of size $l$. We have the following theorem for this example, whose proof can be found in Appendix E.2.

**Theorem E.3** (**Justification example II**). *Suppose each entry of $\theta_\mathcal{T}$ is i.i.d. drawn from the uniform distribution on $[0, 1]$. Let $K = 2$ and $\mathcal{D}_\mathcal{E}, \mathcal{Q}_\mathcal{E}$ be as defined above. For a query $e_q$ drawn from $\mathcal{Q}_\mathcal{E}$, let $L, L'$ denote the expected prediction loss of $e_q$ using* TopK *and* TopK-Div, *respectively, where the randomness comes from $\theta_\mathcal{T}$, $e_q$, and the selection of demonstration examples. Then $L > L'$ if hyperparameter $\alpha \geq 1 - 1/l$ for* TopK-Div.

The demonstration examples of TopK and TopK-Div must cover all non-zero entries of $e_q$. The smaller loss of TopK-Div is caused by selecting two demonstration examples with different non-zero entries among $\{2l + 1, \ldots, 4l\}$, indicating that adding diversity could benefit ICL "beyond coverage".

In Appendix E.3, we conduct simulations to validate that the advantage of TopK-Div over TopK, driven by coverage and beyond, extends to more general settings, including the ID setting ($\mathcal{D}_\mathcal{E} = \mathcal{Q}_\mathcal{E}$) and scenarios with different training scales for $D$.

### E.1 PROOF OF THEOREM E.2: JUSTIFICATION EXAMPLE I

Fix a query $e_q$ drawn from $\mathcal{Q}_\mathcal{E}$. By symmetry, we can assume the non-zero entry set of $e_q$ is $[2l]$. For simplicity, we let $\theta = \theta_\mathcal{T}$.

**Demonstration example set for TopK-Div** We first analyze the demonstration example set for TopK-Div, denoted by $S = \{s^{(1)}, s^{(2)}\} \subseteq D$. Let $T^{(t)}$ denote the non-zero entry set of $s^{(t)}$. By the construction of $\mathcal{D}$, we first note that $|T^{(1)} \cap [l]| = \frac{l}{2}$. By the rule of TopK-Div, we also note that $|T^{(2)} \cap [l]| = \frac{l}{2}$ and $T^{(1)} \cap T^{(2)} = \emptyset$. Such $s^{(2)}$ must exist since all elements in the ground set of $\mathcal{D}_\mathcal{E}$ are contained in $D$, and is selected since it minimizes

$$\alpha \cdot \texttt{Similarity}(e, e_q) + (1 - \alpha)\texttt{Diversity}(e, S)$$

over all $e \in D - \{s^{(1)}\}$.

**Demonstration example set for TopK** Next, we compute the expected prediction loss $L$ for TopK. Again, let its demonstration example set be $S = \{s^{(1)}, s^{(2)}\} \subseteq D$. Let $T^{(t)}$ denote the non-zero

entry set of $s^{(t)}$. By the construction of $\mathcal{D}$, we note that $|T^{(1)} \cap [l]| = |T^{(2)} \cap [l]| = \frac{l}{2}$. However, different from the case of TopK-Div, $|T^{(1)} \cap T^{(2)}|$ can vary from 0 to $l - 1$. To handle this, we define $a = |T^{(1)} \cap T^{(2)} \cap [l]|$ and $b = |T^{(1)} \cap T^{(2)} \cap ([d] \setminus [l])|$, and define $L_{a,b}$ to be the expected prediction loss conditioned on pair $(a, b)$. Note that $0 \leq a, b \leq l/2$ and $a + b \leq l - 1$.

**Comparing $L$ and $L'$**  We remark that $L$ is a linear combination $\sum_{a,b} p_{a,b} L_{a,b}$ with $\sum_{a,b} p_{a,b} = 1$, where $p_{a,b}$ is the conditional probability with respect to intersection numbers $(a, b)$. Also, $L' = L_{0,0}$. By symmetry, we have the following observation:

$$\Pr[a \leq l/4 \leq b] \geq 0.25,$$

where $l/4$ is the expectation of $a$ and $b$. Thus, we have

$$L \geq \sum_{a \leq l/4 \leq b} p_{a,b} L_{a,b} \geq \sum_{a,b \in l/4 \pm \sqrt{l}} p_{a,b} \cdot \min_{a \leq l/4 \leq b} L_{a,b} \geq 0.25 \min_{a \leq l/4 \leq b} L_{a,b}.$$

Thus, to prove $L > L'$, it suffices to prove the following lemma.

**Lemma E.4 (Comparing $L_{a,b}$ and $L_{0,0}$).** *For any $a \leq l/4 \leq b$, we have $L_{a,b} > 4L_{0,0}$.*

*Proof.* By symmetry, we assume $T^{(1)} = [\frac{l}{2}] \cup ([\frac{5}{2}l] - [2l])$, $T^{(2)} = ([l] - [a] - [\frac{l}{2} - a]) \cup ([3l - b] - [\frac{5}{2}l - b])$, $|T^{(1)} \cap T^{(2)} \cap [L]| = |T^{(1)} \cap T^{(2)} \cap [2L]| = a$, $|T^{(1)} \cap T^{(2)} \cap ([4L] - [2L])| = b$. The expected prediction loss for this setting equals $L_{a,b}$ since $\theta_i$s are i.i.d. random variables. Let $\widehat{\theta}$ denote the min-norm solution defined as in Assumption E.1. Then we have

$$\langle \widehat{\theta} - \theta, e_{T^{(1)}} \rangle = \sum_{i=1}^{\frac{l}{2}} \widehat{\theta}_i + \sum_{i=2l+1}^{\frac{5}{2}l} \widehat{\theta}_i - \sum_{i=1}^{\frac{l}{2}} \theta_i - \sum_{i=2l+1}^{\frac{5}{2}l} \theta_i = 0, \tag{5}$$

and

$$\langle \widehat{\theta} - \theta, e_{T^{(2)}} \rangle = \sum_{i=\frac{l}{2}-a+1}^{l-a} \widehat{\theta}_i + \sum_{i=\frac{5}{2}l-b+1}^{3l-b} \widehat{\theta}_i - \sum_{i=\frac{l}{2}-a+1}^{l-a} \theta_i - \sum_{i=\frac{5}{2}l-b+1}^{3l-b} \theta_i = 0. \tag{6}$$

To get the min-norm solution, we need to minimize the following Lagrangian multiplier

$$\mathcal{L}(\widehat{\theta}, \lambda_1, \lambda_2) = \sum_{i=1}^{l-a} \widehat{\theta}_i^2 - 2\lambda_1 \langle \widehat{\theta} - \theta, e_{T^{(1)}} \rangle - 2\lambda_2 \langle \widehat{\theta} - \theta, e_{T^{(2)}} \rangle.$$

To ensure the partial derivatives with respect to $\widehat{\theta}$ equal to 0, we obtain that

$$\begin{aligned}
\widehat{\theta}_1 = \ldots = \widehat{\theta}_{\frac{l}{2}-a} = \widehat{\theta}_{2l+1} = \ldots = \widehat{\theta}_{\frac{5}{2}l-b} &= \lambda_1, \\
\widehat{\theta}_{\frac{l}{2}+1} = \ldots = \widehat{\theta}_{l-a} = \widehat{\theta}_{\frac{5}{2}l+1} = \ldots = \widehat{\theta}_{3l-b} &= \lambda_2, \\
\widehat{\theta}_{\frac{l}{2}-a+1} = \ldots = \widehat{\theta}_{\frac{l}{2}} = \widehat{\theta}_{\frac{5}{2}l-b+1} = \ldots = \widehat{\theta}_{\frac{5}{2}l} &= \lambda_1 + \lambda_2, \\
\widehat{\theta}_{l-a+1} = \ldots = \widehat{\theta}_{2l} = \widehat{\theta}_{3l-b+1} = \ldots = \widehat{\theta}_{4l} &= 0.
\end{aligned} \tag{7}$$

Adding Equations (5)-(7), we have

$$(l + a + b)(\lambda_1 + \lambda_2) = \sum_{i=1}^{\frac{l}{2}} \theta_i + \sum_{i=2l+1}^{\frac{5}{2}l} \theta_i + \sum_{i=\frac{l}{2}-a+1}^{l-a} \theta_i + \sum_{i=\frac{5}{2}l-b}^{3l-b} \theta_i. \tag{8}$$

Thus, we conclude that

$$\left[\sum_{i=1}^{l}\widehat{\theta}_i - \sum_{i=1}^{l}\theta_i\right]^2$$

$$= \left[(\frac{l}{2} - a)(\lambda_1 + \lambda_2) + a\lambda_1 + a\lambda_2 - \sum_{i=1}^{l}\theta_i\right]^2$$

$$= \left[\frac{l}{2}(\lambda_1 + \lambda_2) - \sum_{i=1}^{l}\theta_i\right]^2$$

$$= \left[\frac{\frac{l}{2}}{l+a+b}\left(\sum_{i=1}^{\frac{l}{2}}\theta_i + \sum_{i=2l+1}^{\frac{5}{2}l}\theta_i + \sum_{i=\frac{l}{2}-a+1}^{l-a}\theta_i + \sum_{i=\frac{5}{2}l-b}^{3l-b}\theta_i\right) - \sum_{i=1}^{l}\theta_i\right]^2$$

$$= \left[-\frac{\frac{l}{2}+a+b}{l+a+b}\sum_{i=1}^{\frac{l}{2}-a}\theta_i - \frac{\frac{l}{2}+a+b}{l+a+b}\sum_{i=\frac{l}{2}+1}^{l-a}\theta_i - \frac{a+b}{l+a+b}\sum_{i=\frac{l}{2}-a+1}^{\frac{l}{2}}\theta_i\right.$$

$$\left. + \frac{\frac{l}{2}}{l+a+b}\sum_{i=2l+1}^{\frac{5}{2}l}\theta_i + \frac{\frac{l}{2}}{l+a+b}\sum_{i=\frac{5}{2}l-b+1}^{3l-b}\theta_i\right]^2,$$

where the first equation follows from Equation (7) and the third equation follows from Equation (8). Since each $\theta_i$ is i.i.d. drawn from the uniform distribution over $[0, 1]$, we have

$$L_{a,b} = \mathbb{E}\left[\langle\widehat{\theta} - \theta, e_q\rangle^2\right]$$

$$= \mathbb{E}\left[\left[\sum_{i=1}^{l}\widehat{\theta}_i - \sum_{i=1}^{l}\theta_i\right]^2\right]$$

$$= \frac{(\frac{l}{2}+a+b)^2(\frac{l}{2}-a) + \frac{a(a+b)^2}{2} + \frac{l^3}{8} + \frac{3(bl-a^2-ab)^2}{2}}{6(l+a+b)^2}.$$

Thus, $L_{0,0} = \frac{l}{24}$. When $a \leq l/4 \leq b$, we have

$$L_{a,b} > \frac{3(bl - a^2 - ab)^2/2}{6(l+a+b)^2}$$

$$\geq \frac{(l^2/4 - 2(l/4)^2)^2}{4(2l)^2} \qquad (a \leq l/4 \leq b)$$

$$= \frac{(l^2/8)^2}{16l^2}$$

$$= \frac{l^2}{1024}$$

$$\geq 4L_{0,0}. \qquad (l \geq 200)$$

This completes the proof. $\qquad\qquad\qquad\qquad\qquad\qquad\qquad\qquad\qquad\qquad\qquad\qquad\qquad\square$

### E.2 PROOF OF THEOREM E.3: JUSTIFICATION EXAMPLE II

By symmetric, we fix $e_q = e_{[l]}$. Like the proof of Theorem E.2, we first study the demonstration example sets, denoted by $S = \{s^{(1)}, s^{(2)}\} \subseteq D$, derived from TopK and TopK-Div. We observe that for both algorithms, $|T^{(1)} \cap [l]| = |T^{(2)} \cap [l]| = l - 1$. Note that this property for TopK-Div follows from the choice of $\alpha \geq 1 - \frac{1}{l}$, which ensures that $|T^{(2)} \cap [l]| \leq l - 2$ can not achieve the minimum for

$$\alpha \cdot \texttt{Similarity}(e, e_q) + (1 - \alpha)\texttt{Diversity}(e, S)$$

Thus, by symmetry, we can fix $T^{(1)} = [l-1] \cup \{2l+1]\}$ and there are only three choices for $T^{(2)}$:

- Case 1: $T^{(2)} = [l] \cup \{2l + 2\} - \{1\}$;
- Case 2: $T^{(2)} = [l] \cup \{2l + 1\} - \{1\}$.
- Case 3: $T^{(2)} = [l - 1] \cup \{2l + 2\}$.

We define the expected prediction loss of these three cases to be $L_1, L_2, L_3$, respectively. By the definition of TopK-Div, we know that $L' = L_1$. Moreover, the expected prediction loss $L$ of TopK must be a linear combination of $L_1, L_2, L_3$. Thus, it suffices to prove that $L_2 > L_1$ and $L_3 > L_1$. Below, we compute $L_1, L_2, L_3$ separately.

**Computing $L_1$.** The computation idea is similar to that of Lemma E.4. Suppose $\widehat{\theta}$ is the min-norm solution and we have

$$\sum_{i=1}^{l-1} \widehat{\theta}_i + \widehat{\theta}_{2l+1} - \sum_{i=1}^{l-1} \theta_i - \theta_{2l+1} = 0 \text{ and } \sum_{i=2}^{l} \widehat{\theta}_i + \widehat{\theta}_{2l+2} - \sum_{i=2}^{l} \theta_i - \theta_{2l+2} = 0.$$

Again, consider the Lagrangian multiplier $\mathcal{L}(\widehat{\theta}, \lambda_1, \lambda_2) = \sum_{i=1}^{l} (\widehat{\theta}_i)^2 - 2\lambda_1 \langle \widehat{\theta} - \theta, e_{T^{(1)}} \rangle - 2\lambda_2 \langle \widehat{\theta} - \theta, e_{T^{(2)}} \rangle$. To ensure the partial derivative w.r.t. $\widehat{\theta}$ equal to 0, we have

$$\widehat{\theta}_2 = \widehat{\theta}_3 = \ldots = \widehat{\theta}_{l-1} = \lambda_1 + \lambda_2, \text{ and } \widehat{\theta}_1 = \widehat{\theta}_{2l+1} = \lambda_1, \widehat{\theta}_l = \widehat{\theta}_{2l+2} = \lambda_2.$$

Combining the above equations, we have

$$(2l - 2)(\lambda_1 + \lambda_2) = 2\sum_{i=2}^{l-1} \theta_i + \theta_1 + \theta_l + \theta_{2l+1} + \theta_{2l+2}.$$

Thus,

$$(\sum_{i=1}^{l} \theta_i - \sum_{i=1}^{l} \widehat{\theta}_i)^2 = [(l-1)(\lambda_1 + \lambda_2) - \sum_{i=1}^{l} \theta_i]^2 = (\frac{\theta_{2l+1} + \theta_{2l+2} - \theta_1 - \theta_l}{2})^2.$$

Consequently, we have

$$L_1 = \mathbb{E}[\langle \widehat{\theta} - \theta, e_q \rangle^2] = \mathbb{E}[(\frac{\theta_{2l+1} + \theta_{2l+2} - \theta_1 - \theta_l}{2})^2] = \frac{1}{12}.$$

**Computing $L_2$.** Similarly, we have

$$\sum_{i=1}^{l-1} \widehat{\theta}_i + \widehat{\theta}_{2l+1} - \sum_{i=1}^{l-1} \theta_i - \theta_{2l+1} = 0, \text{ and } \sum_{i=2}^{l} \widehat{\theta}_i + \widehat{\theta}_{2l+1} - \sum_{i=2}^{l} \theta_i - \theta_{2l+1} = 0.$$

Thus, using the Lagrangian multiplier, we obtain that

$$\widehat{\theta}_2 = \widehat{\theta}_3 = \ldots = \widehat{\theta}_{l-1} = \widehat{\theta}_{2l+1} = \lambda_1 + \lambda_2, \text{ and } \widehat{\theta}_1 = \lambda_1, \widehat{\theta}_l = \lambda_2.$$

Combining the above equations, we have

$$(2l - 1)(\lambda_1 + \lambda_2) = 2\sum_{i=2}^{l-1} \theta_i + 2\theta_{2l+1} + \theta_1 + \theta_l.$$

Thus,

$$\begin{aligned} L_2 = & \ \mathbb{E}[(\sum_{i=1}^{l} \theta_i - \sum_{i=1}^{l} \widehat{\theta}_i)^2] = \mathbb{E}[\sum_{i=1}^{l} \theta_i - [(l-1)(\lambda_1 + \lambda_2)]^2] \\ = & \ \mathbb{E}[\frac{1}{2l-1} \sum_{i=2}^{l-1} \theta_i + \frac{l}{2l-1}(\theta_1 + \theta_l) - \frac{2l-2}{2l-1} \theta_{2l+1}]^2] = \frac{9l^2 - 7l + 2}{12(12l-1)^2} > L_1. \end{aligned}$$

**Computing $L_3$.**   Similarly, we have

$$\sum_{i=1}^{l-1} \widehat{\theta}_i + \widehat{\theta}_{2l+1} - \sum_{i=1}^{l-1} \theta_i - \theta_{2l+1} = 0, \text{ and } \sum_{i=1}^{l-1} \widehat{\theta}_i + \widehat{\theta}_{2l+2} - \sum_{i=1}^{l-1} \theta_i - \theta_{2l+2} = 0.$$

Using the Lagrangian multiplier, we obtain that

$$\widehat{\theta}_1 = \widehat{\theta}_2 = \widehat{\theta}_3 = \ldots = \widehat{\theta}_{l-1} = \lambda_1 + \lambda_2, \text{ and } \widehat{\theta}_{2l+1} = \lambda_1, \widehat{\theta}_{2l+2} = \lambda_2.$$

Combining the above equations, we have

$$(2l - 1)(\lambda_1 + \lambda_2) = 2\sum_{i=1}^{l-1} \theta_i + \theta_{2l+1} + \theta_{2l+2}.$$

Thus,

$$
\begin{aligned}
L_3 = & \ \mathbb{E}[(\sum_{i=1}^{l} \theta_i - \sum_{i=1}^{l} \widehat{\theta}_i)^2] = \mathbb{E}[\sum_{i=1}^{l} \theta_i - [(l-1)(\lambda_1 + \lambda_2)]^2] \\
= & \ \mathbb{E}[\frac{1}{2l-1} \sum_{i=1}^{l-1} \theta_i + \theta_l - \frac{l-1}{2l-1}(\theta_{2l+1} + \theta_{2l+2})]^2] = \frac{9l^2 - 7l + 2}{12(2l-1)^2} > L_1.
\end{aligned}
$$

Overall, we complete the proof of Theorem E.3.

### E.3   EXPERIMENT SETTINGS

We consider the ID setting with $\mathcal{D}_\mathcal{E} = \mathcal{Q}_\mathcal{E}$.

**Metric for coverage.**   Given a sample $(e, y_E)$, let $T^{(e)}$ denote the non-zero entry set of $e$. Given a demonstration example set $S \subseteq D$ and a query $e_q$, we define the coverage ratio of $S$ with respect to $e_q$ to be:

$$r_S(e_q) := \frac{|(\bigcup_{e \in S} T^{(e)}) \cap T^{(e_q)}|}{|T^{(e_q)}|},$$

i.e., the ratio of non-zero entries of $e_q$ covered by samples in $S$. By definition, $r_S(e_q) \in [0, 1]$ and a larger $r_S(e_q)$ represents higher coverage. Specifically, when $r_S(e_q) = 1$, we say $e_q$ is fully covered by $S$. Moreover, given a method $\mathcal{A}$ that generates a demonstration example set $A(e_q) \subseteq D$ for each query $e_q$, we define

$$r(\mathcal{A}) := \mathbb{E}_{e_q \sim \mathcal{Q}_\mathcal{E}}[r_{\mathcal{A}(e_q)}(e_q)] \tag{9}$$

to be the expected value of its coverage ratio $r_{\mathcal{A}(e_q)}(e_q)$. If $r(\mathcal{A}) = 1$, we say every query is fully covered by $\mathcal{A}$.

We want to study the loss difference between `TopK-Div` and `TopK` under two scenarios: 1) when query $e_q$ is fully covered by both algorithms `TopK-Div` and `TopK`, i.e., $r(\text{TopK-Div}) = r(\text{TopK}) = 1$; and 2) when the coverage ratio of `TopK-Div` is smaller than that of `TopK`, i.e., $r(\text{TopK-Div}) < r(\text{TopK})$.

**Parameters.**   Let $d = 200$. Let $l$ vary from $3, 4, 8$. Let $K = 4$ or $8$. Let $\mathcal{D}_\mathcal{E} = \mathcal{Q}_\mathcal{E}$ be the distribution that first samples a subset $T \subset [d]$ of size $l$ and then generate $e_T$. We set the size of training set $D$ to be $|D| = d \times \text{train\_scale}$, where $\text{train\_scale} \in \{1, 5, 10\}$.

For each pair $(l, K)$, we generate a testing set $D_{\text{test}}$ of size 100. We ensure that $D_{\text{test}} \cap D = \emptyset$. We report the expected prediction loss and coverage ratio of `TopK` and `TopK-Div` for each pair $(l, K)$.

### E.4   RESULT AND DISCUSSIONS

The results, reported in Table 19, reveal key insights into the performance differences between `TopK` and `TopK-Div`. We observe that when $l = 8$, the coverage ratio of `TopK` is lower than

Table 19: (**Simulation of the min-norm solution**) "Coverage" represents the coverage ratio of methods, defined as in Equation (9). For each random seed, we selected one hundred test samples. We report the average results across 3 different random seeds for each metric.

| Method | Shot | Metric | Train scale = 1 | | | Train scale = 5 | | | Train scale = 10 | | |
|---|---|---|---|---|---|---|---|---|---|---|---|
| | | | $l=3$ | $l=4$ | $l=8$ | $l=3$ | $l=4$ | $l=8$ | $l=3$ | $l=4$ | $l=8$ |
| TopK | $K=4$ | Loss | 0.21 | 0.31 | 12.70 | 0.15 | 0.30 | 9.55 | 0.19 | 0.30 | 7.51 |
| | | Coverage | 1.00 | 1.00 | 0.55 | 1.00 | 1.00 | 0.61 | 1.00 | 1.00 | 0.66 |
| | $K=8$ | Loss | 0.47 | 0.57 | 3.09 | 0.43 | 0.84 | 1.33 | 0.45 | 0.83 | 1.19 |
| | | Coverage | 1.00 | 1.00 | 0.75 | 1.00 | 1.00 | 0.80 | 1.00 | 1.00 | 0.81 |
| TopK-Div | $K=4$ | Loss | 0.19 | 0.32 | 10.25 | 0.18 | 0.31 | 5.47 | 0.21 | 0.29 | 3.97 |
| | | Coverage | 1.00 | 1.00 | 0.63 | 1.00 | 1.00 | 0.75 | 1.00 | 1.00 | 0.80 |
| | $K=8$ | Loss | 0.31 | 0.38 | 2.58 | 0.23 | 0.38 | 1.32 | 0.20 | 0.38 | 1.75 |
| | | Coverage | 1.00 | 1.00 | 0.87 | 1.00 | 1.00 | 0.94 | 1.00 | 1.00 | 0.94 |

that of TopK-Div, while its loss is significantly higher. For example, when $l=8$, $K=4$, and train_scale = 5, the coverage ratio is $r(\text{TopK}) = 0.61$, compared to $r(\text{TopK-Div}) = 0.75$, while the loss for TopK is 9.55, notably larger than the 5.47 observed for TopK-Div. This demonstrates that incorporating diversity can reduce prediction loss by improving coverage, aligning with Theorem E.2.

When $l=3$ or $4$, the coverage ratios of TopK and TopK-Div are both 1. We find that the loss of TopK is comparable to or even lower than that of TopK-Div when $K=4$, but significantly higher when $K=8$, across various training scales. For instance, when $l=3$, $K=8$, and train_scale = 5, the loss for TopK is 0.43, whereas for TopK-Div it is 0.23. This supports our findings in Theorem E.3, demonstrating that diversity can enhance in-context learning beyond just coverage. The inverse trend in loss between $K=4$ and $K=8$ suggests that increasing coverage is beneficial when the query is not fully covered but becomes redundant when the demonstration example set already provides sufficient coverage.

## THE USE OF LARGE LANGUAGE MODELS (LLMS)

We employed large language models (LLMs) solely for word-level grammar checking and minor stylistic refinement of the manuscript. Beyond this limited function, LLMs did not contribute to any other aspects of our research or writing, including conceptualization, experimental design, data analysis, or interpretation of results.

