# OpenReview forum: "[ICLR 2026] The role of diversity in in-context learning for large language models"
_ICLR.cc/2026/Conference — ICLR 2026 Conference Withdrawn Submission_

### Official Review · Reviewer_ZTFq · 2025-10-19

**Soundness:** 2
**Presentation:** 3
**Contribution:** 2
**Rating:** 2
**Confidence:** 4

**Summary:**

This paper explores in-context learning (ICL) abilities of large language models (LLMs), focusing on the effect of demonstration diversity on downstream tasks. Specifically, authors conducted experiments on 5 tasks and 3 open-source LLMs and claimed three findings: diversity matters for 'challenging' questions.

**Strengths:**

- This paper is well-written and easy to follow.
- The research topic of this paper is interesting.

**Weaknesses:**

1. Authors claim in the abstract  that 'the impact of diversity in example selection remains underexplored. '  However,  there has been many works exploring the diversity in ICL demonstration both theoretically and empirically. For example,  [1-5].

2. The paper's central claim relies on the notion that diversity helps on "harder" tasks, but "difficulty" is never rigorously defined. The proxy used in Table 3 (fine-tuning-based classification) is limited to two datasets and doesn't generalize. In other words, this claim is rather subjective instead of proved quantitatively.

3. The distinction between coverage-based benefits and "beyond coverage" effects is central to the paper's contribution, but the evidence is insufficient. Section 3.2 and Appendix C.5 provide limited experiments on only SQuAD with data perturbations.

4. The theoretical justification (Theorems E.2, E.3) uses toy linear regression with binary embeddings and assumes min-norm solutions (Assumption E.1). The gap between these idealized settings and real language tasks with continuous embeddings and complex LLM behaviors is vast. Thus, this is questionable.

5. Many reported improvements are <1% (e.g., Table 7 shows average Δ of 0.27% for simple tasks). Given the standard deviations in Table 10 (often 0.3-0.6%), many differences may not be statistically significant.

6. While the paper tests different α values (TopK-Div) and subset sizes (Div), the analysis is fragmented across appendices without clear prescriptive guidelines. Practitioners need to know how to set these hyperparameters.

7. On SST-2, TopK consistently outperforms diversity methods by 1-2% (Table 1), but the paper dismisses this as "simple tasks prefer TopK" without deeper analysis.

8. Appendix D.5 shows that method rankings can change with different embeddings (BM25 vs. BertScore vs. Sentence-BERT), suggesting findings are highly embedding-dependent.

9. One major concern is that Diversity-aware methods require computing pairwise similarities or running greedy selection algorithms, which have higher computational costs than simple TopK. This is never discussed.

10. Table 2 shows mixed OOD results. For sentiment (SST-2 to IMDB), TopK-Div helps; but for commonsense reasoning (CsQA to ARC-Easy), the benefit is inconsistent (0.4% to 1.0% depending on model).

11. The paper uses a specific diversity formulation (Equation 2: 1 - average similarity), but other diversity metrics exist (e.g., determinantal point processes, MaxMin diversity, entropy-based measures). Further diversity metrics exploration are needed.

12. While Appendix D.2 shows results for varying K, the main paper only tests k in {4,8}. The relationship between diversity benefits and K is complex (Figure 6 shows non-monotonic trends).

13. Most experiments use open-source models from three families (Llama, Gemma, Mistral). Only two commercial models tested briefly in Appendix D.1 (Table 14). Results may not generalize to other model families or architectures.

14. The paper claims novelty in studying diversity vs. coverage-based methods, but doesn't clearly delineate how TopK-Div and Div differ from these prior coverage-focused approaches beyond conceptual level.


References

[1]Diverse Demonstrations Improve In-context Compositional Generalization https://aclanthology.org/2023.acl-long.78.pdf

[2]Exploring the Role of Diversity in Example Selection for In-Context Learning https://arxiv.org/abs/2505.01842

[3]Enhancing Contrastive Demonstration Selection with Semantic Diversity for Robust In-Context Machine Translation

[4]In-Context Learning with Iterative Demonstration Selectionhttps://arxiv.org/abs/2504.09305

[5]Affinity and Diversity: A Unified Metric for Demonstration Selection via Internal Representations https://arxiv.org/abs/2502.14380

**Questions:**

Please see weaknesses.

---

### Official Review · Reviewer_DtJJ · 2025-10-26

**Soundness:** 3
**Presentation:** 3
**Contribution:** 2
**Rating:** 6
**Confidence:** 3

**Summary:**

This paper investigates how diversity in demonstration selection impacts in-context learning (ICL) for large language models. It introduces four selection methods: Rand (random), TopK (most similar examples), Div (diverse coreset with similarity), and TopK-Div (a method balancing similarity and diversity). Theoretical analysis shows that diversity improves both coverage and generalization. Experiments are conducted across tasks including sentiment classification, commonsense reasoning, math, code, and reading comprehension, using models like Llama-3, Gemma-2, and Mistral. Results show that diversity-aware methods, especially TopK-Div, significantly improve performance on complex and out-of-distribution tasks.

**Strengths:**

1. The proposed TopK-Div method offers a flexible trade-off between similarity and diversity, making it practical and easily adaptable.

2. It provides formal theoretical analysis explaining why diversity helps in ICL, including scenarios where it goes beyond simple feature coverage.

3. The paper conducts extensive experiments across a wide range of tasks (classification, reasoning, generation) and models (Llama-3, Gemma-2, Mistral), demonstrating the robustness of the findings.

**Weaknesses:**

1. The assumptions in the analysis framework are strong, which may not clearly reflect the true underlying mechanism of the in-contex learning.

2. The downstream tasks used in the paper are relatively simple, making it difficult to demonstrate whether the proposed method is applicable to real-world scenarios—for instance, the industry has already adopted challenging datasets such as AIME.

3. More discussion about diversity and related baselines(see Question2-4).

**Questions:**

1. Is there any method here that can quickly determine α based on the characteristics of the dataset and the model?

2. The effectiveness of this method depends on the performance of the embedding model; if the embedding model fails to effectively capture the information in the data, the resulting diversity will be inaccurate. Could data diversity be considered from more dimensions? (keywords, gradient or other domain-related metrics)

3. Following Question 2, it is worth investigating what "diversity" truly means in the context of In-Context Learning (ICL). If diversity is assessed solely based on embeddings, it appears to be unrelated to the final model performance (e.g., Llama-3.1, Gemma-2, and Mistral-v0.3). Would model-aware metrics—such as those based on gradients—provide a better measure of diversity?

4. Moreover, the idea of jointly considering diversity and similarity has already been addressed in ICML 2025 by [1], which employs coreset [2] and optimal transport. Could you clarify the similarities and differences between your method and that of [1]?

[1] Coresets for Data-efficient Training of Machine Learning Models
[2] Tarot: Targeted data selection via optimal transport

---

### Official Review · Reviewer_JBs8 · 2025-10-30

**Soundness:** 2
**Presentation:** 3
**Contribution:** 2
**Rating:** 2
**Confidence:** 4

**Summary:**

This work investigates the impact of diversity in example selection for in-context learning (ICL) in large language models (LLMs). The authors analyze five example selection methods: Rand, TopK, Div-S3, Div, and TopK-Div. A theoretical framework is introduced to explain how diversity enhances coverage and generalization. Through a series of experiments across various NLP tasks, including sentiment classification, commonsense reasoning, math, code, and reading comprehension, the authors demonstrate that diversity-aware methods—especially TopK-Div—outperform traditional methods like Top-K. The experiments show that these methods improve performance on complex and out-of-distribution tasks, supporting the idea that diversity is crucial for effective in-context learning.

**Strengths:**

1.	The paper is easy to follow overall.

2.	It presents robust and convincing findings supported by extensive and detailed experiments across multiple tasks (classification, reasoning, and generation) and models (Llama-3, Gemma-2, and Mistral).

3.	The work also provides preliminary theoretical analysis and proofs to support and explain the empirical observations.

**Weaknesses:**

1.	The assumptions made in the analytical framework appear rather strong and may not accurately capture the true underlying mechanisms of in-context learning.

2.	The performance gaps between different methods are not very pronounced, and the errors introduced during the evaluation process may lead to performance differences.

**Questions:**

1.	Line 1015-1018: The authors mention, "we compute the logit for 'great' and 'terrible' respectively, and predict the sentiment to be positive if the logit for 'great' is larger than that for 'terrible'."What is the rationale behind this evaluation processing? Why were 'great' and 'terrible' selected instead of 'positive' and 'negative'?

2.	Line 1019-1021: The authors state, "we pick the option with the smallest average cross-entropy loss." What is the basis for this evaluation processing? Why is it preferred over the evaluation processing method used in the classification tasks?

3.	Line 260-261: The authors mention using an instruction-tuned model for math-related tasks, but a base model is used for other tasks. What is the difference between these two models? Would using the instruction-tuned model for open-ended generation tasks be more aligned with real-world applications?

4.	Line 1597-1598: Could the authors clarify the meaning and role of the index $j$ in $S = {(e_{j_i}, y_{j_i})}^K_{i=1}$?

5.	Kapuriya et al. [1] have investigated a nearly identical research topic to the one presented in this paper. I believe this may be a crucial reference that has not been discussed.
[1] Kapuriya, Janak, et al. "Exploring the role of diversity in example selection for in-context learning." Proceedings of the 48th International ACM SIGIR Conference on Research and Development in Information Retrieval. 2025.

---

### Official Review · Reviewer_EFVC · 2025-11-01

**Soundness:** 3
**Presentation:** 3
**Contribution:** 2
**Rating:** 4
**Confidence:** 3

**Summary:**

This paper presents a systematic study on the role of diversity in in-context learning (ICL) for large language models (LLMs). While previous works focus primarily on selecting demonstrations most similar to the query, this work investigates whether and when diverse examples improve ICL performance. The authors evaluate several diversity-aware selection strategies—Div, Div-S3, and TopK-Div—across five task categories (classification, reasoning, math, code generation, reading comprehension) and three major open-source model families (LLaMA-3.1, Gemma-2, Mistral-v0.3).

**Strengths:**

1. This is one of the first comprehensive studies to quantify the impact of diversity in retrieval-based ICL across various models and task types.

2. Extensive experiments across multiple datasets and model scales (1B–70B) support the claims, with consistent results.

3. The paper provides actionable insights—e.g., use diversity for complex or OOD settings and similarity for simple tasks.

4. The inclusion of baselines (TopK, Rand, Div-S3) and the introduction of the tunable TopK-Div strategy are well-motivated.

**Weaknesses:**

1. Although a theoretical framework is mentioned, it remains underdeveloped and placed mostly in the appendix. The main text lacks rigorous formalization or proofs explaining why diversity helps beyond coverage.

2. The diversity methods (Div, TopK-Div) are adaptations of existing techniques (e.g., DPP, submodular selection). The contribution lies more in systematic evaluation than methodological innovation.

3. The experiments focus on English text tasks only. The generality of the findings to multimodal, multilingual, or conversational settings is untested.

4. The reported improvements (1–3% on difficult tasks) are statistically small, though consistent. A discussion of variance and significance would strengthen the argument.

5. The paper could benefit from qualitative examples showing how diverse demonstrations change model reasoning patterns or attention distributions.

**Questions:**

1. Strengthen the theoretical analysis in the main text—possibly connecting diversity to representation coverage, mutual information, or implicit gradient-based learning in ICL.

2. Include statistical tests (e.g., paired t-tests) to demonstrate the significance of performance differences.

3. Provide qualitative or visualization-based analysis showing how diversity changes the model’s behavior (e.g., activation patterns, attention focus).

4. Discuss computational cost implications of diversity-aware selection methods.

5. Explore cross-lingual or domain-specific tasks (e.g., medical, legal) to validate generality.

6. Compare against recent retrieval-enhanced ICL methods (e.g., Dr.ICL, RePrompt, or reinforcement-based retrievers).

---

### Note · Authors · 2025-11-17

I have read and agree with the venue's withdrawal policy on behalf of myself and my co-authors.